# Strong degradation of palsas and peat plateaus in northern Norway during the last 60 years

Amund F. Borge, Sebastian Westermann, Ingvild Solheim, Bernd Etzelmüller

Department of Geosciences, University of Oslo, P.O. Box 1047, 0316 Oslo, Norway

*Correspondence to*: S. Westermann (sebastian.westermann@geo.uio.no)

**Abstract.** Palsas and peat plateaus are permafrost landforms occurring in subarctic mires which constitute sensitive ecosystems with strong significance for vegetation, wildlife, hydrology and carbon cycle. We have systematically mapped the occurrence of palsas and peat plateaus in the northernmost county of Norway (Finnmark, ~50,000 km$^2$) by manual interpretation of aerial images from 2005-2014 at a spatial resolution of 250 m$^2$. At this resolution, mires and wetlands with

palsas or peat plateaus occur in about 850 km$^2$ of Finnmark, with the actual palsas and peat plateaus underlain by permafrost covering a surface area of approximately 110 km$^2$. Secondly, we have quantified the lateral changes of the extent of palsas and peat plateaus for four study areas located along a NW-SE transect through Finnmark by utilizing repeat aerial imagery from the 1950s to the 2010s. The results of the lateral changes reveal a total decrease of 33-71 % in the areal extent of palsas and peat plateaus during the study period, with the largest lateral change rates observed in the last decade. However, the

results indicate that degradation of palsas and peat plateaus in northern Norway has been a consistent process during the second half of the 20[th] century and possibly even earlier. Significant rates of areal change are observed in all investigated time periods since the 1950s, and thermokarst landforms observed on aerial images from the 1950s suggest that lateral degradation was already an ongoing process at this time. The results of this study show that lateral erosion of palsas and peat plateaus is an important pathway for permafrost degradation in the sporadic permafrost zone in northern Scandinavia. While

the environmental factors governing the rate of erosion are not yet fully understood, we note a moderate increase in both air temperature, precipitation and snow depth during the last few decades in the region.

## 1 Introduction

Palsas and peat plateaus are the most common landforms indicating permafrost in Fennoscandia, with widespread abundance in the sporadic permafrost zone in the northern region of Norway, Finland and Sweden (Seppälä, 1986). Palsas and peat

plateaus are subarctic permafrost landforms in mires defined as *"a peaty permafrost mound possessing a core of alternating layers of segregated ice and peat or mineral soil material"* and as *"a generally flat-topped expanse of peat, elevated above the general surface of a peatland, and containing segregated ice that may or may not extend downward into the underlying mineral soil"* (van Everdingen, 1998). Palsa mires demarcate the outer or lower limit for permafrost in a given area (Sollid and Sørbel, 1998) and are found in a narrow climatic envelope (Parviainen and Luoto, 2007). The permafrost in palsas is

thus relatively warm, with a mean annual ground temperature (MAGT) often close to 0 °C in northern Fennoscandia (Christiansen et al., 2010; Johansson et al., 2011). Thus, palsas are vulnerable to climate change (e.g. Aalto et al., 2014).

Palsas in northern Fennoscandia have been extensively studied in terms of processes and controlling factors (e.g. Kujala et al., 2008; Seppälä, 2011; Seppälä and Kujala, 2009), distribution (e.g. Luoto et al., 2004a; Malmström, 1988; Meier, 1996; Sollid and Sørbel, 1998) and stratigraphy and dating (e.g. Oksanen, 2006; Seppälä, 2005; Vorren, 1972). In 2004, the Norwegian institute for nature research (NINA) started a surveillance program of palsa mires in Norway, monitoring six selected palsa mires in Norway (Hofgaard, 2004). However, a recent inventory of palsa mire and peat plateaus and their long-term changes is lacking, despite earlier mapping approaches (Meier, 1987; Sollid and Sørbel, 1998, 1974).

In the Nordic countries, warming of permafrost in mountain areas has been evident since the beginning of 2000 (Christiansen et al., 2010; Isaksen et al., 2011; Isaksen et al., 2007), and degradation of sporadic permafrost in palsa mires has been observed at sites in northern Sweden (Zuidhoff and Kolstrup, 2000), southern Norway (Matthews et al., 1997; Sollid and Sørbel, 1998, 1974) and northern Norway (Hofgaard and Myklebost, 2015, 2014; Meier and Thannheiser, 2015).

Palsa mires and peat plateaus occur in many areas of the subarctic, e.g. in northwestern and northeastern Canada (Lewkowicz et al., 2011; Payette et al., 2004), in European Russia (Kuhry and Turunen, 2006) and in western Siberia (Blyakharchuk and Sulerzhitsky, 1999). The palsa mires in Fennoscandia constitute the westernmost edge of the discontinuous and sporadic lowland permafrost zone in northwestern Russia, where peat plateaus and palsa mires are abundant features (e.g. Väliranta et al., 2003). Ground temperatures observed in peat plateaus in Fennoscandia are generally higher than in Russia (Mazhitova et al., 2004; Sannel et al., 2015), so that changes observed in Fennoscandia today may be a window to the future development of the much larger areas in Russia.

Palsa mires constitute a sensitive ecosystem, with a biologically distinct and heterogeneous environment (Luoto et al., 2004b). Moreover, studies from Russia show that palsas and peat plateaus can contain a significant fraction of soil organic carbon within a landscape (Hugelius et al., 2011). An increase in carbon fluxes (especially $CH_4$) to the atmosphere from the thawing organic-rich permafrost soils could turn the subarctic to a net carbon source (Koven et al., 2011; Schaefer et al., 2011). Furthermore, significant emissions of the greenhouse gas $N_2O$ have been observed in peat plateaus in northwestern Russia and in palsa mires in Finland (Marushchak et al., 2011; Repo et al., 2009), highlighting the active role of palsas and peat plateaus in the coupled carbon and nitrogen cycles. In river hydrology, several studies suggest permafrost thawing as one contributing reason to a widely observed increase in river base flow (e.g. Bense et al., 2012; St Jacques and Sauchyn, 2009; Walvoord and Striegl, 2007), which has also been observed in areas of Fennoscandia where palsa mires are known to exist (Sjöberg et al., 2013; Wilson et al., 2010).

In this study, we provide a quantitative assessment of the current extent of palsas and peat plateaus in Finnmark, the northernmost county of Norway. Furthermore, we quantify the degradation of selected palsa mires and peat plateaus due to lateral erosion between the 1950s until today. By "degradation", we refer to the processes (or the result of these processes) that decrease the volume of palsas and peat plateaus. With "lateral erosion", we mean the lateral decrease in size (as seen on

2D aerial imagery) of palsas and peat plateaus, where the margin of palsas or peat plateaus is transformed to wetland. Lateral erosion is often due to block erosion, but may also be a result of ground subsidence due to melting of excess ground ice at the edge, followed by submergence below the water table of the surrounding wet mire.

## 2 Setting

The county of Finnmark is situated in northern Norway between roughly 68° N and 71° N and features a land area of 48 618 km$^2$ (Fig. 1). The geomorphology of Finnmark is dominated by alpine mountains in the northwest, and the plateau-like landscape of Finnmarksvidda at an elevation of about 300-500 m a.s.l. in the interior and south. The area was covered by ice sheets several times during the Pleistocene, and the plateau is normally covered by thick cover of ground moraines and glacio-fluvial and glacio-lacustrine sediments (Sollid et al., 1973). Wetlands and mires fill the depressions between moraines and ridges.

The climate in Finnmark is influenced by the North Atlantic Current, and varies from a relatively wet and warm maritime climate at the coast in the northwest to a rather dry and cold environment in the more continental Finnmarksvidda (Vikhamar-Schuler et al., 2010). The average winter (December-February) and summer (June-August) temperature (1961-1990) for the plateau are around -15 °C and 10 °C, respectively, with mean annual air temperature (MAAT, 1961-1990) generally varying between -2 °C and -4 °C (Aune, 1993). In comparison, the coast has a MAAT (1961-1990) mostly above 0 °C (Aune, 1993). The mean annual precipitation (MAP, 1961-1990) ranges from more than 1000 mm at the coast to less than 400 mm on Finnmarksvidda, while the mean annual (hydrological year) maximum snow depth (MASD, 1971-2000) ranges from less than 50 cm on Finnmarksvidda to more than 200 cm at the outer coast (seNorge, 2016). On Finnmarksvidda, the mean annual number of days with dry snow (MADDS, 1961-1990) is generally between 150 and 200 (seNorge, 2016), and the mean fraction of snow of the total precipitation (MSFr, 1961-1990) is usually less than 40 % (seNorge, 2016).

According to regional modelling approaches at 1 km$^2$ spatial resolution, permafrost underlays about one fifth of the land surface in Finnmark (Farbrot et al., 2013; Gisnås et al., 2013). The altitude of the lower limit of discontinuous mountain permafrost is above c. 500 m a.s.l. in continental areas of Finnmark (Farbrot et al., 2013), whereas sporadic permafrost in palsa mires can exist almost down to sea level.

Four study areas roughly situated in a NE-SW transect through Finnmark are chosen: *Karlebotn*, *Lakselv*, *Suossjavri*, and *Goatheluoppal* (Fig. 1), with climate data (seNorge, 2016) provided in Table 2. Two of the study areas, *Karlebotn* (25-40 m a.s.l.) and *Lakselv* (15-70 m a.s.l.), are located in a maritime setting below the local marine limit. The mires in *Karlebotn* (70°23′ N, 28°25′ E) are located close to inner parts of Varangerfjorden, eastern Finnmark, consisting of extensive peat plateaus and disintegrated remains of these. The mires in *Lakselv* (70°4′ N, 25°3′ E) are located in the inner parts of Porsangerfjorden, near inhabited areas belonging to the small town of Lakselv. Here, the mires consist of some larger peat plateaus or disintegrated remains of peat plateaus, and some larger thermokarst lakes. *Suossjavri* (300-350 m

a.s.l., 69°23′ N, 24°15′ E) is situated on the plateau of Finnmarksvidda, with mires featuring both peat plateaus and dome palsas. *Goatheluoppal* (440 m a.s.l., 68°54′ N, 22°21′ E) is located in a flat area at the southwestern edge of Finnmarksvidda, approximately 5 km from the border of Finland. The mires in this area consist of dome palsas, which are small in extent but easily visible on aerial images. Larger peat plateaus do not exist here. Fig. 2 shows two examples of degradation and disappearance of palsas from *Suossjavri* and *Karlebotn*.

## 3 Methodology and data

In a first step, we map the presence of palsas and peat plateaus using a grid-based approach with a spatial resolution of 250 $m^2$ for the entire county of Finnmark. The occurrence of palsas and peat plateaus is based on visual interpretation of orthorectified aerial images provided on Norgeibilder.no by the Norwegian Mapping Authority (Norgeibilder, 2015) at 1:4000 to 1:6000 (which is less than the best possible resolution). The occurrence of palsas/peat plateaus is assigned to grid-cells that include one or more palsas with a diameter of at least 10 m. The threshold is defined for practical reasons, as a palsa with diameter of less than 10 m is difficult to detect and correctly interpret at the scale employed for the mapping due to the quality of the aerial images.

The resulting map of palsa/peat plateau distribution is evaluated against gridded datasets of MAAT (seNorge, 2016) for the normal period 1961-1990 and a Digital Elevation Model (DEM) (Kartverket, 2015b) at 10 $m^2$ resolution. The MAAT dataset originally features a spatial resolution of 1 $km^2$ and has been resampled to 250 $m^2$ resolution using bilinear resampling technique to compare with the palsa/peat plateau map. As mires generally are situated in topographic depressions, we use the minimum elevation in each 250 $m^2$ grid-cells to assess the altitudinal distribution of palsa mires and peat plateaus.

To estimate the total area of the actual palsas and peat plateaus in Finnmark, we multiply the surface area of 250 $m^2$ grid-cells with the presence of palsas by a mean areal fraction that is covered by palsas/peat plateaus in each grid cell. To obtain an estimate for the latter, we delineate the exact boundaries of palsas and peat plateaus from the most recent available images (2005-2012) at the four study areas, covering in total 260 grid-cells, which we assume to be a representative sample for Finnmark. This mapping is performed at the best possible resolution (c. 1:1250), so that also palsas with diameter smaller than 10 m are included (for grid cells for which palsa/peat plateau occurrence is determined by the 250 m scale mapping). All digital geodata handling is performed in ArcGIS (© ESRI, Redlands, CA, USA).

To ensure a consistent interpretation of the extent of palsas on the aerial images, the same person delineated the palsas for each individual study area. To estimate the accuracy of the manual and thus to a certain extent subjective delineation process, parts of the Karlebotn study area (~ 0.24-0.26 $km^2$) were independently mapped (using the images from 2008) by two persons. This comparison yielded a difference of 8 % in the total area which can be considered a rough estimate for the mapping accuracy.

To determine the uncertainty induced by the 10 m threshold in the 250 m scale mapping (see above), we once again investigate the four main study areas and their surroundings (c. 140 km$^2$). By mapping at best possible resolution, palsas with smaller diameter can be detected which facilitates estimating the number of 250 m grid-cells excluded due to the mapping threshold. We find that the number of grid-cells with presence of palsas is 8.6 % higher when including palsas and palsa remnants with diameters less than 10 m. However, the total area covered by palsas/peat plateaus increases by only 0.16 % due to the limited area of these palsas. We therefore conclude that our mapping can provide a robust estimate for the total area covered by palsas/peat plateaus, although isolated small palsas occur regularly in grid cells flagged as free of palsas/peat plateaus.

In a second step, changes of palsa and peat plateau extent from the 1950s to now are evaluated for the four study areas. Here, aerial images at 1:20 000 to 1:50 000 scale with a ground sample distance (GSD) ranging from 0.24-0.50 m (Table 1) are extracted by the following procedures: (1) Aerial images from 2003 and 2008-2012 are directly provided on Norgeibilder.no. (2) Analogue aerial images from 1956-1959 and 1980-1982 are scanned (© The Norwegian Mapping Authority, Norway) and georeferenced. Due to the nature of palsas being situated in flat mires, georeferencing is considered to give sufficient accuracy in this study. The georeferencing procedure for *Lakselv*, *Suossjavri* and *Goatheluoppal* is concentrated around the individual mires, yielding low RMSEs for the investigated areas, mainly around 0.5-2 m. The two images from 1957 of *Karlebotn* are georeferenced using control points within the whole scene, resulting in RMSEs of 5.7 and 8.7 m. For the study areas *Lakselv* and *Karlebotn*, only images from the 1950s and 2005/2008 are utilized.

Polygons that match the individual palsas are produced by visual interpretation from the different time slices (example in Fig. 3). Knowledge obtained from fieldwork in *Suossjavri* in summer 2014 was crucial in the process to recognize and separate palsas from the rest of the landscape. Palsas and peat plateaus situated at the edge of the mire in sloping terrain covered by moraine material are especially difficult to delineate, as there is often a diffuse transition between the moraines and the palsa area. To avoid an effect on the change detection, the same boundaries are chosen for all time slices in such areas. For simplicity, thermokarst features within palsas or peat plateaus smaller than c. 10 m, e.g. small drainages and depressions with or without ponds, are ignored in the delineation process, as they are likely to be still underlain by permafrost (Sjöberg et al., 2015).

Finally, the observed changes are evaluated against changes in climatic variables. We use gridded data at a spatial resolution of 1 km$^2$ of air temperature, precipitation and maximum snow depth (during a hydrological year) provided by the Norwegian Meteorological Institute (Mohr, 2008; Mohr and Tveito, 2008; Tveito and Førland, 1999) and the Norwegian Water Resources and Energy Directorate through www.senorge.no (seNorge, 2016). With the seNorge (2016) dataset, we calculate MAAT, MAP, MASD, MADDS and MSFr at the location of the four study areas. For each location, a trend analysis is performed using linear regression.

The seNorge (2016) data set of air temperature and precipitation is based on interpolation between measurements at meteorological stations using altitudinal scaling (Mohr and Tveito, 2008). Maximum snow depth, days of dry snow and snow fraction of the total precipitation are based on a snow model using the gridded air temperature and precipitation as

input forcing (Saloranta, 2016). While the seNorge (2016) dataset is a consistent, gap-free dataset available for all study areas, the sparsity of meteorological stations in Finnmark leads to higher uncertainties on Finnmarksvidda compared with densely populated areas in south of Norway (Tveito et al., 2000). In addition, the occurrence of strong winter inversions in mire areas in northern Fennoscandia (e.g. Nordli, 1990; Pike et al., 2013) hampers interpolating meteorological observations.

## 4 Results

### 4.1 The distribution of palsas and peat plateaus in Finnmark

Palsas or peat plateaus are identified in 13 752 grid cells of 250 m size which corresponds to a total area of about 850 km$^2$ (Fig. 4). The distribution map shows that palsas are scattered over most of the more continental parts of Finnmark, with the highest concentration in the inner parts of Finnmarksvidda towards the southern border to Finland (Fig. 4). In the southeastern corner of Finnmarksvidda, close to the Finnish border, the abundance of palsa mires and peat plateaus is significantly lower than in the more central parts. The coastal sites of *Lakselv* and *Karlebotn* represent areas with medium to high concentrations of palsas. The dominating elevation range of palsa occurrence is around 300-500 m a.s.l., corresponding to the common elevations in Finnmarksvidda (Fig. 5a). In addition, a significant number of palsas and peat plateaus occur at low elevations of 0-100 m a.s.l., mainly in coastal areas. Palsas and peat plateaus are most concentrated at MAATs between -3 °C and -4 °C (Fig. 5b), but at a few sites they occur at locations with MAAT as high as +1 °C.

High-resolution delineation of palsas and peat plateaus in the four study areas (for the 2010s) covered in total 260 grid cells, corresponding to about 2 % of the 250 m grid cells with palsas/peat plateaus in Finnmark. The sites cover a gradient of climatic and environmental conditions across Finnmark, so that we consider the results a plausible first-order estimate, although it is unclear if the four sites are a fully representative subsample. We find a total area of 2.13 km$^2$ within the 260 grid cells, yielding an average areal fraction of palsas/peat plateaus of about 13 % in grid cells with presence of these features. The present-day total area of palsas and peat plateaus in Finnmark can thus be estimated to about 110 km$^2$ or 0.2 % of the total land area of Finnmark, with an estimated uncertainty on the order of 10 km$^2$ due to the manual delineation process (Sect. 3).

### 4.2 Areal change through lateral erosion

In the four study areas, a total area of 4.4 km$^2$ of palsas and peat plateaus were mapped by aerial images from the 1950s, which was reduced to less than half (2.13 km$^2$) in the 2010s. No new palsas with diameter of more than 10 m were found and no palsas increased visibly in extent between the end of 1950s and 2010s. We note that the reduction in areal extent is significantly larger than the estimated accuracy of the manual delineation process (8 % of the total mapped area, Sect. 3).

Data from seNorge (2016) indicate that all of the four study areas have experienced a notable increase in MAAT during the study period (Table 2), while MAP increased slightly at two sites and remained more or less constant at the others. MASD increased in all areas except Lakselv according to seNorge (2016) data, although it is unclear if this result is

representative for palsas and peat plateaus, as snow depths on palsas/peat plateaus are generally much lower than in the surrounding wet mire due to wind redistribution, as e.g. observed in Suossjavri and Lakselv in March 2013.

*Karlebotn* – Two large peat plateau complexes bordered by palsa mires were mapped (Fig. 6a). The total area for all palsas and peat plateaus was 2.17 km$^2$ in 1957, while it was reduced to 1.0 km$^2$ in 2005/2008, corresponding to a 54 % decrease in areal extent during 50 years (Fig. 6b). The mean annual rate of reduction based on the original area from the 1950s (from now on referred to as the mean annual loss rate) was on the order of 1 % a$^{-1}$. The *Karlebotn* site features the highest increase in MAAT during the study period, with a MAAT close to 0 °C for the last decades (Table 2).

*Lakselv* – In total eight distinct palsa mires were investigated in the *Lakselv* area (Fig. 7a). On average, they featured a decrease of 48 % in the total area of palsas and peat plateaus from 1959 to 2008, corresponding to a decrease from 0.95 km$^2$ to 0.49 km$^2$ (Fig. 7b) and a mean annual loss rate of about 1.0 % a$^{-1}$. In the *Lakselv* area, some of the mires are surrounded by cropland, and the image analysis revealed that small parts of palsa mires 5-8, including a few palsas, have been transformed to cropland during the period 1959-2008. However, this anthropogenic effect is only responsible for around 1% of palsa area that was lost in the 50-year study period, so that its influence on the mean annual loss rates is negligible. At the *Lakselv* sites, the MAAT is above 0 °C (Table 2), thus placing these palsa mires at the extreme upper end of the temperature range within which palsa mires are found (Fig. 5b). The meteorological station at Banak (5 m a.s.l.), situated at the coast a few kilometres away from the palsa mires, has documented a high average wind speed of about 5 m/s (based on data from 1967-2014, MET, 2015) and frequent wintertime rain events are known to occur (e.g. Vikhamar-Schuler et al., 2010; Vikhamar-Schuler et al., 2013). Both factors could lead to a reduction of the insulating wintertime snow cover on the palsas and thus influence the ground thermal regime, possibly contributing to the thermal stability of permafrost in the palsas. This interpretation is corroborated by observations during a field visit in March 2013, where the palsas were either completely free of snow or covered by an ice layer, while a snow cover exceeding 0.5 m depth was observed in the surrounding birch forest.

*Suossjavri* – In total seven distinct mires with palsas and peat plateaus were mapped (Fig. 8a). Around a third of the original area of palsas and peat plateaus in 1956/1959 had degraded by 2011 (Fig. 8b and 9, Table 3). The mean annual loss rates were relatively stable in time, with a notable acceleration in the most recent time period (Fig. 9). The larger peat plateaus in the *Suossjavri* area featured a smaller annual loss rate compared to smaller palsas. From 1956/1959 and 2011, the area of the four largest peat plateaus was reduced by 15 %, compared to 48 % areal loss for palsas and smaller peat plateaus. Measurements at the meteorological station Cuovddatmohkki (286 m a.s.l.), located around 5-10 km from the *Suossjavri* palsa mires, suggest a slightly colder and dryer environment than the seNorge (2016) data set (Table 2), with a MAAT of -2.6 °C for the normal period 1961-1990 (Aune, 1993) and a MAP of 380 mm for the same period (Førland, 1993). With about 1.5 m/s, the average wind speed at Cuovddatmohkki (based on data from 1967-2014, MET, 2015) is significantly lower than in the *Lakselv* area.

*Goatheluoppal* – The area features a large number of smaller palsas distributed within a large wetland and mire complex (Fig. 10a). Although the MAAT is rather cold at the site (Table 2), it experienced the by far largest changes in areal

extent (Fig. 10b and 11, Table 4): in 2012, 71 % of the original palsa area from 1958 had disappeared. The mean annual loss rates were remarkably stable at this study area (Fig. 11), with an average rate of 1.3 % $a^{-1}$. According to seNorge (2016), *Goatheluoppal* featured the highest increase in both MAP and MASD of all study areas (Table 2).

If one extrapolates the mean observed annual loss rate of around 1 % $a^{-1}$ to the entire county of Finnmark, about half of the area covered by palsas and peat plateaus in the 1950s has disappeared until today, corresponding to a total area loss of 110 $km^2$. In the earliest available images from the 1950s, landforms indicative of palsa degradation are already visible at all study areas. Examples are thermokarst lakes and rim ridges visible in the aerial images from 1958 from *Goatheluoppal* which most likely mark the position and the extent of former palsas (Fig. 12). This interpretation is corroborated by their

similarity to degradational landforms in the 2012 images which were still mature palsas in 1958. The analysis of the aerial images from the 1950s therefore indicates that palsa degradation was already an ongoing process at this time.

## 5 Discussion

### 5.1 Comparison to previous studies

The mapped distribution of palsas and peat plateaus (Fig. 4a) is largely in agreement with the less detailed palsa mire map by

Sollid and Sørbel (1998) (Fig. 4b) which is mostly based on work from the 1960s-1970s (Sollid and Sørbel, 1974). A notable difference is the presence of palsa mires in the Pasvik valley in the southeastern corner of Finnmark in the earlier map. In our study (Fig. 4a), no palsas were detected in this area, but large mire areas with evidence of former palsas were observed. This suggests that palsas have disappeared entirely from some marginal areas of the present distribution in the past 50 years.

In northern Fennoscandia, degradation of palsas has been documented in eastern Finnmark (Hofgaard and

Myklebost, 2014), Troms county (Hofgaard and Myklebost, 2015) and in the southernmost palsa mire in northern Sweden (Zuidhoff and Kolstrup, 2000). However, palsa mires at the southern margin of the palsa extent on the Kola Peninsula, Russia, displayed no change in both areal extent, thickness of the active layer and height and number of permafrost mounds over the past 80 years (Barcan, 2010). Furthermore, investigations in the wider *Goatheluoppal* area (southeast of the mires mapped in this study) revealed no overall change in the distribution and size of palsas from 2004 to 2011 (Hofgaard and

Myklebost, 2012). These conflicting results suggest that local factors – and not only the regional climate – influence the thermal regime in the ground and thus the survival of palsas and peat plateaus. Such factors could be snow redistribution due to wind drift (Seppälä, 1982), the thickness of peat and its water content (Kujala et al., 2008), groundwater flow (e.g. Vallee and Payette, 2007; Thórhallsdóttir, 1994) and vegetation (Zuidhoff and Kolstrup, 2005).

The palsa distribution map of Finnmark represents all palsas/peat plateaus that are well visible in aerial images.

However, isolated small palsas (with a diameter of less than 10 m) are not well recognizable so that they are not contained in the map. A more detailed assessment in the four study areas suggests that the total number of 250 m grid cells with palsas

and peat plateaus may be up to 10 % higher if also isolated small palsas are included (Sect. 3). However, as these unmapped permafrost features are very small, their contribution to the total area is negligible.

The total area covered by palsas/peat plateaus has been computed from the gridded 250 m palsa distribution map using an average grid cell fraction that was determined by manual delineation of the palsa/peat plateau boundaries in four study areas covering about 2 % of the total number of grid cells containing palsas/peat plateaus. The manual mapping is associated with errors, e.g. by subjectively defining the palsa margins. This "human" error is estimated to be on the order of 10 % from independent mapping of two persons (Sect. 3), which can provide a rough estimate for the grid cell fraction and the hereof computed total area covered by palsas/peat plateaus. Finally, it is unclear whether the four study areas are fully representative for the entire region, although they are situated along a transect spanning a wide range of conditions under which palsas/peat plateaus occur in Finnmark.

## 5.2 Palsas and peat plateaus in northern Norway

Dating of peat layers in palsas in northern Europe and northwestern Russia reveals that some palsas and peat plateaus were formed during cold conditions in the Little Ice Age (LIA), while others are significantly older and survived the Medieval warm period (Oksanen, 2005). While similar dating studies are lacking for our study area, the few datings available for northern Finland and Sweden (overview in Sannel and Kuhry, 2011) suggest that at least a large part of the palsa mires and peat plateaus in Finnmark is of LIA origin. Thus, it is well possible that these features entered a stage of slow, but consistent degradation in conjunction with post-LIA warming which is still ongoing today. At the two sites with repeat aerial imagery (*Goatheluoppal* and *Suossjavri*), the mean annual loss rates were remarkably stable over the past 50 years and do not seem to be directly correlated to MAAT, although the accelerated degradation at *Suossjavri* in the past decades may partly be related to pronounced increases of air temperatures since the 1980s (Hanssen-Bauer, 2005). The mean annual loss rates found in this study are of similar order of magnitude as rates observed in other areas, for example in northern Sweden (Zuidhoff and Kolstrup, 2000) and in the discontinuous permafrost zone in northwest (Kershaw, 2003; Quinton et al., 2011) and eastern Canada (Payette et al., 2004; Vallee and Payette, 2007).

Temperature projections indicate that Finnmarksvidda could experience a significant temperature increase during this century: for the moderate future emission scenario RCP4.5 (IPCC, 2014), an increase in MAAT of 3.6 °C from the period 1971-2000 to the period 2071-2100 is estimated (Hanssen-Bauer et al., 2015). Thus, only about one third of the palsas and peat plateaus will be situated in areas with MAAT below 0 °C by the end of this century (see Fig. 5b) so that the degradation is likely to continue or even accelerate in the future. Even without future acceleration, palsas and peat plateaus at the investigated areas will have disappeared by 2030 (*Goatheluoppal*), 2060 (*Lakselv* and *Karlebotn*) and 2080 (*Suossjavri*), if the mean annual loss rates observed in this study continue. On the other hand, the existence of palsas in coastal areas with MAAT above 0 °C leaves the possibility open that a small portion of these permafrost landforms may survive also in a future climate, so that a definite conclusion on the fate of palsas and peat plateaus in Finnmark in the 21[st] century cannot be drawn yet.

## 5.3 Degradation of palsas and peat plateaus through lateral erosion

Palsas and peat plateaus are complex permafrost features, and the environmental factors governing their formation and degradation are not yet fully understood. Since all stages of palsa development can be found in the same mire, Seppälä (1982, 1986) suggested that changes in climate are not necessarily the reason for the collapse of individual palsas, but a natural part of the cyclic development of palsas. Results of studies by Zoltai (1993) and Matthews et al. (1997) support this view. In contrast, we detected neither new palsas larger than 10 m diameter nor palsas that increased in areal extent in the investigated 60-year-period, although detection of small new embryo palsas can be difficult in aerial images. Other studies confirm this general pattern of degradation of palsas (Hofgaard and Myklebost, 2015, 2014; Sollid and Sørbel, 1998; Zuidhoff and Kolstrup, 2000), or present evidence of a wider abundance of palsas in the past (Luoto and Seppälä, 2003) which is an indication that the climatic factors are a primary control on the distribution of palsas and peat plateaus. The two different views are not necessarily in conflict, as both processes can be important for different time-periods and for different spatial scales (Aalto and Luoto, 2014). Cyclical evolution of palsas as observed by Seppälä (1982, 1986) could e.g. occur for a certain range of climatic conditions, while aggradation or degradation could be dominant during colder and warmer periods, respectively.

In addition to climatic factors, the mean areal loss rates appear to be related to shape and geometry of individual palsas and peat plateaus. The results in *Suossjavri* show a clear difference between the mean annual loss rate for large peat plateaus and smaller palsas, with smaller palsas degrading significantly faster than larger peat plateaus. A potential explanation is that dome palsas are higher than peat plateaus, resulting in more block erosion, while low peat plateaus are mostly eroded by thermal erosion from water (Sollid and Sørbel, 1998). However, many of the delineated palsas are probably "residual" palsas from the disintegration of larger peat plateaus and are therefore not necessarily higher than the adjacent peat plateaus, while still featuring large degradation rates. The explanation may thus simply be that smaller and more fractionated features have a longer perimeter on which lateral erosion can act relative to their area. If the absolute rates of lateral retreat (in m a$^{-1}$) are on average of similar magnitude for all features, the areal loss rate (in % a$^{-1}$) would indeed be larger for smaller palsas, in agreement with the findings of this study.

Fig. 13 shows an aerial image of a peat plateau near *Suossjavri* highly affected by block erosion, as common for palsas and peat plateaus in this area. At the actively degrading margins, the mire vegetation is not yet established and a water-filled depression forms, indicating that the retreat of the margin occurs at higher velocity than the regrowing of the mire vegetation. However, the water bodies become overgrown and many of them eventually disappear which is evident from both the aerial images and field observations. The proximity between the standing water and the ice-rich core of the peat plateaus and palsas most likely contributes to thermal undercutting and eventually block erosion at the margins (Kurylyk et al., 2016), but a variety of factors, such as the height of the palsa and the ground ice content can be expected to play a role for this process.

On the other hand, the interior of palsas and peat plateaus can also experience thaw subsidence resulting in thermokarst depressions and suprapermafrost taliks, as seen for peat plateaus in northern Sweden (Åkerman and Johansson, 2008, Sjöberg et al., 2015). Based on calculated thaw rates and an instant increase in air temperature of 2 °C, Sjöberg et al. (2015) estimated that it will take 175-260 years for the permafrost at their investigated peat plateaus to completely thaw. However, much more rapid degradation has been observed in the same region (Zuidhoff, 2002), which could be an indication that lateral erosion considerably increases the degradation rates. A recent study in south-central Alaska found that 85 % of the degradation of forested permafrost plateaus was due to lateral degradation along the margins (Jones et al., 2016).

## 5.4 Implications for permafrost modelling and mapping

Grid-based modelling of the ground thermal state suggests that several thousand square kilometres in Finnmark could be covered by permafrost in mires areas (Farbrot et al., 2013; Gisnås et al., 2013; Gisnås et al., 2016), which is significantly more than the area of palsas and peat plateaus of 110 km$^2$ estimated in this study. However, the model results are based on 1 km$^2$ grid cells without accounting for the considerable subgrid variability of the ground thermal regime. In this study, on average only 13 % of the area of grid cells of 250 m$^2$ size (i.e. 16 times smaller than 1 km$^2$ grid cells) proved to be actually covered by clear permafrost features, and this fraction is necessarily even lower for 1 km$^2$ grid cells. Therefore, it seems possible to console at least the orders of magnitude obtained with the two approaches: the modelling result refers to the area with climatic conditions that make the presence of permafrost and thus palsas and peat plateaus possible, if further conditions e.g. related to the ground thermal properties (Farbrot et al., 2013; Gisnås et al., 2013) are met. The latter, however, only occur for a rather small fraction within each grid cell where palsas and peat plateaus are actually observed in aerial images in this study. Although it is possible that permafrost in mire areas in Finnmark can also exist outside of palsas and peat plateaus, we suggest using the lower number of 110 km$^2$ as a conservative estimate for the permafrost area in mires in Finnmark in the future. We conclude that it is not possible to model the occurrence and thermal regime of palsa and peat plateaus in grid-based approaches without taking subgrid variability of environmental factors explicitly into account. There exist a variety of approaches how such small-scale variability of different factors can be included in modelling (e.g. Fiddes et al., 2015; Kurylyk et al., 2016; Westermann et al., 2015; Zhang et al., 2012). As exemplified by Gisnås et al. (2014) for mountain permafrost environments in Norway, redistribution of snow due to wind drift could be a governing factor for the ground thermal regime also in palsa mires, especially since palsas and peat plateaus are elevated landscape elements which feature lower snow depths than the surrounding mire area (Seppälä, 1982). Tiling approaches that facilitate a statistical representation of subgrid variability of snow depths have been implemented in a regional permafrost model for the Norwegian mountain areas (Gisnås et al., 2015), but the concept has not yet been applied to mire areas.

Up to now, a physically-based model approach for the evolution of palsas and peat plateaus is lacking, which is not surprising considering that the environmental factors giving rise to their formation, stability and degradation are not fully understood yet (Sect. 5.1). Changes in vegetation and surface properties accompanying the disappearance of palsas and peat

plateaus (Fig. 2) further complicate modelling efforts. This study is clear evidence that lateral erosion is a dominant pathway for degradation of permafrost features in the sporadic permafrost zone in northern Norway. On the other hand, it is unclear if the areal loss has been accompanied by systematic elevation changes of the surface due to melting of excess ground ice in the palsas and peat plateaus, followed by the drainage of the meltwater. One-dimensional model approaches for ground subsidence and thermokarst pond formation (Lee et al., 2014; Westermann et al., 2015) are a first step towards physically-based modelling of thaw processes in ice-rich permafrost ground, but they must be included in a model framework that facilitates representing small-scale redistribution of heat, water and snow. Furthermore, fully coupled 3D-models have been demonstrated for larger thermokarst lakes (Kessler et al., 2012), and similar concepts may also be applicable to model palsa and peat plateau dynamics. We conclude that this study constitutes a solid baseline that can help to guide and validate future model improvements regarding peat plateaus and palsas.

## 6 Conclusion

Using high-resolution (0.2-0.5 $m^2$) aerial imagery, we systematically map the occurrence of palsas and peat plateaus on 250 m grids in the sporadic permafrost zone in northern Norway. Furthermore, we delineate the exact boundaries of palsas and peat plateaus at four study areas along a NE-SW transect covering both coastal and more continental inland areas (in total about 2% of the palsa mires and peat plateaus in northern Norway). Using repeat aerial images from the 1950s to the 2010s, changes in areal extent over time are investigated.

- We estimate that about 110 $km^2$ of the county of Finnmark (corresponding to c. 0.2 % of the land surface) are currently covered by palsas and peat plateaus, with an estimated uncertainty of about 10 $km^2$. The largest concentrations occur in the plains in the inner parts of Finnmark at elevations of 300 to 500 m a.s.l. and mean annual air temperatures between -3 and -4 °C. However, palsas/peat plateaus also exist in coastal areas below 100 m a.s.l. with mean annual air temperatures as high as +1 °C.

- Since the 1950s, the area covered by palsas and peat plateaus steadily decreased at all the four study areas through lateral erosion and formation of thermokarst ponds, with a total decrease in area between 33 % and 71 %. Newly formed palsas of diameter of more than 10 m were not observed in the study areas. In the same period, air temperatures increased by 1.0 to 1.5 °C in the study areas.

- Combining the change rates with the areal mapping, we estimate that roughly half of the area covered by palsas and peat plateaus in the 1950s in northern Norway, i.e. an area on the order of 100 $km^2$, has disappeared in the last 50 years.

- Signs of degradation, such as thermokarst lakes, observed in the earliest available aerial images suggest that the degradation of palsas and peat plateaus was already an ongoing process in the 1950s.

- In two study areas, the change rates can be resolved in three different time periods covering 10 to 25 years each. The mean annual loss rate is remarkably constant over the last 50 years, although a moderate acceleration during

this period is observed at one of the sites. However, a direct correlation of change rates with increasing air temperatures, precipitation or snow depth does not exist.

The study is evidence of the highly dynamic evolution that subarctic permafrost environments can experience in relatively short time periods. Such fast changes are related to the melting of excess ground ice in palsas and peat plateaus triggering changes of microtopography that result in significant modifications of e.g. vegetation, hydrology and the carbon cycle. Model projections on the future state of such permafrost environments in general cannot account for the underlying physical processes and must therefore be regarded as highly uncertain. However, if the present-day loss rates continue in the future, palsas and peat plateaus will largely disappear in northern Norway in the course of the 21$^{st}$ century.

*Acknowledgements.* This study was funded by the Department of Geosciences, University of Oslo, COUP (project no. 244903/E10; JPI Climate; Research Council of Norway), CRYOMET (project no. 214465, Research Council of Norway), SatPerm (project no. 239918; Research Council of Norway) and PERMANOR (project no. 255331, Research Council of Norway). Thanks to Trond Eiken for help with gathering of aerial images and the process of georeferencing, and to Kenneth Bahr for assistance during fieldwork summer 2014. We thank two anonymous reviewers for valuable suggestions and feedback to an earlier version of this paper. Furthermore, we thank Jess Joar Andersen from the Norwegian Water Resources and Energy Directorate for assistance with the seNorge snow and precipitation data.

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

**Table 1.** Key information about the aerial images utilized in this study. The images are either gathered from Norgeibilder (2015) or ordered directly from the Norwegian Mapping Authority, Norway, through an aerial image archive (Kartverket, 2015a). GSD: ground sample distance, i.e. the distance between the centres of adjacent pixels measured on the ground.

| | Date | Source | Image scale | GSD, m | Film type |
|---|---|---|---|---|---|
| *Karlebotn* | 28.07.1957 | Aerial image archive* | 1:20 000 | 0.42 | Panchromatic (black-white) |
| | 04.07.2005 | Norgeibilder.no | ND | 0.5 | Analog RGB |
| | 19.08.2008 | Norgeibilder.no | ND | 0.5 | Digital RGB |
| *Lakselv* | 20.07.1959 | Aerial image archive* | 1:20 000 | 0.26 | Panchromatic (black-white) |
| | 11.09.2008 | Norgeibilder.no | ND | 0.50 | Digital RGB |
| *Suossjavri* | 16.08/08.09 - 1956 | Aerial image archive* | 1:20 000 | 0.24 | Panchromatic (black-white) |
| | 21.07.1959 | Aerial image archive* | 1:20 000 | 0.24 | Panchromatic (black-white) |
| | 01.07/15.07 - 1982 | Aerial image archive* | 1:25 000 | 0.36 | Analog RGB |
| | 01.07.2003 | Norgeibilder.no | ND | 0.5 | Panchromatic (black-white) |
| | 17.08.2011 | Norgeibilder.no | ND | 0.4 | Digital RGB |
| *Goatheluoppal* | 21-22.08.1958 | Aerial image archive* | 1:20 000 | 0.24 | Panchromatic (black-white) |
| | 18-20.07.1980 | Aerial image archive* | 1:25 000 | 0.31 | Analog RGB |
| | 01.07.2003 | Norgeibilder.no | ND | 0.5 | Panchromatic (black-white) |
| | 14.08.2012 | Norgeibilder.no | ND | 0.4 | Digital RGB |

* http://159.162.103.4/geovekst/Flybildearkiv/ (© Norwegian Mapping Authority, Norway).

**Table 2.** Data of mean annual air temperature (MAAT), mean annual precipitation (MAP), mean annual maximum snow depth (MASD), mean annual days of dry snow (MADDS) and mean snow fraction of total precipitation (MSFr) during the period 1961-1990 and 1991-2014 at   the four study areas, including linear trends for MAAT, MAP and MASD based on simple linear regression. The data are based on the seNorge (2016) dataset.

| | MAAT | | | MAP | | | MASD | | | MADDS | MSFr |
|---|---|---|---|---|---|---|---|---|---|---|---|
| | 1961-1990 (°C) | 1991-2014 (°C) | Trend (°C(10a)$^{-1}$) | 1961-1990 (mm) | 1991-2014 (mm) | Trend (%(10a)$^{-1}$) | 1961-1990 (mm) | 1991-2014 (mm) | Trend (%(10a)$^{-1}$) | 1961-1990 (days) | 1961-1990 (%) |
| *Karlebotn* | -1.3 | -0.1 | +0.34 | 440 | 477 | +2.6 | 644 | 718 | +16.8 | 150 | 40 |
| *Lakselv* | +0.1 | +1.1 | +0.25 | 377 | 392 | +0.6 | 436 | 401 | -24.7 | 140 | 35 |
| *Suossjavri* | -1.9 | -0.9 | +0.23 | 492 | 510 | +0.2 | 593 | 711 | +12.1 | 186 | 37 |
| *Goatheluoppal* | -2.9 | -1.8 | +0.32 | 354 | 428 | +5.8 | 488 | 599 | +34.5 | 187 | 39 |

**Table 3.** Area of all palsas and peat plateaus in *Suossjavri* for 1956/1959, 1982, 2003 and 2011, with total differences of the
area between 1956/1959 and 2011.

|  | Total all palsas |
| --- | --- |
| **1956/1959 (m$^2$)** | 739817 |
| **1982 (m$^2$)** | 648695 |
| **2003 (m$^2$)** | 553342 |
| **2011 (m$^2$)** | 494507 |
| **Total difference (m$^2$)** | **-245310** |
| **Total difference (%)** | **-33** |

**Table 4.** Area of all palsas in *Goatheluoppal* for 1958, 1980, 2003 and 2012, with total differences of the area between 1958 and 2012.

|  | Total all palsas |
| --- | --- |
| **1958 (m$^2$)** | 501659 |
| **1980 (m$^2$)** | 348973 |
| **2003 (m$^2$)** | 212879 |
| **2012 (m$^2$)** | 146834 |
| **Total difference (m$^2$)** | **-354825** |
| **Total difference (%)** | **-71** |

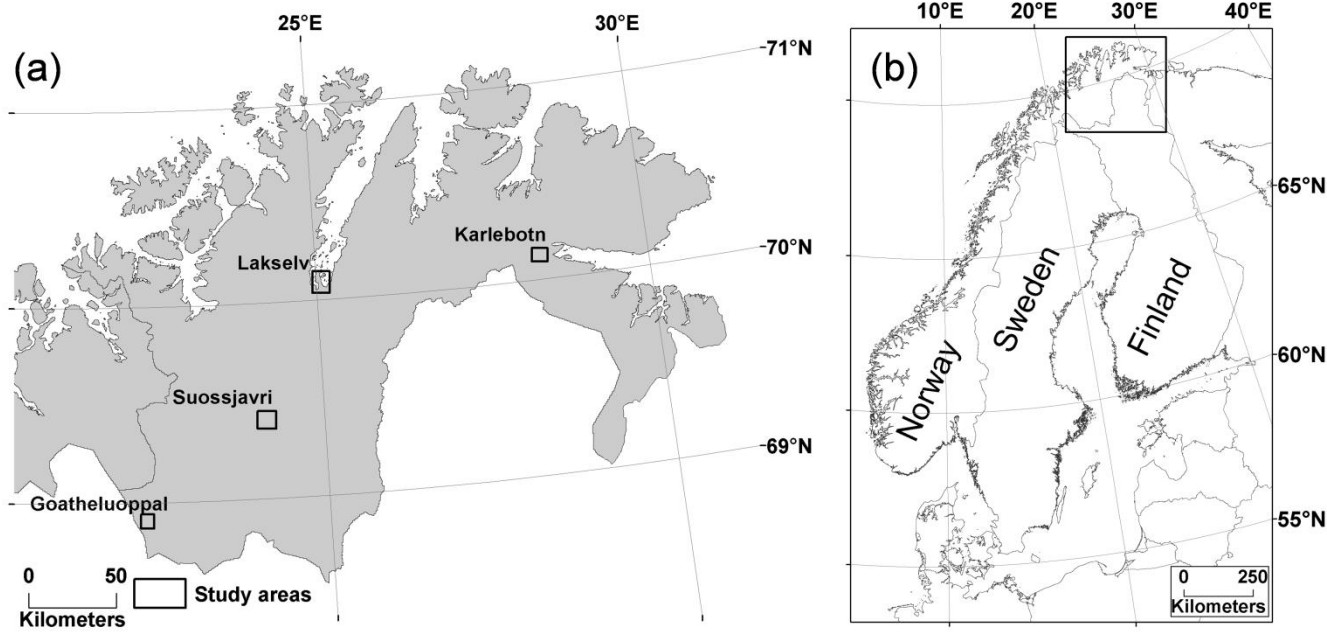

10    **Figure 1.** The county of Finnmark with the four study areas marked.

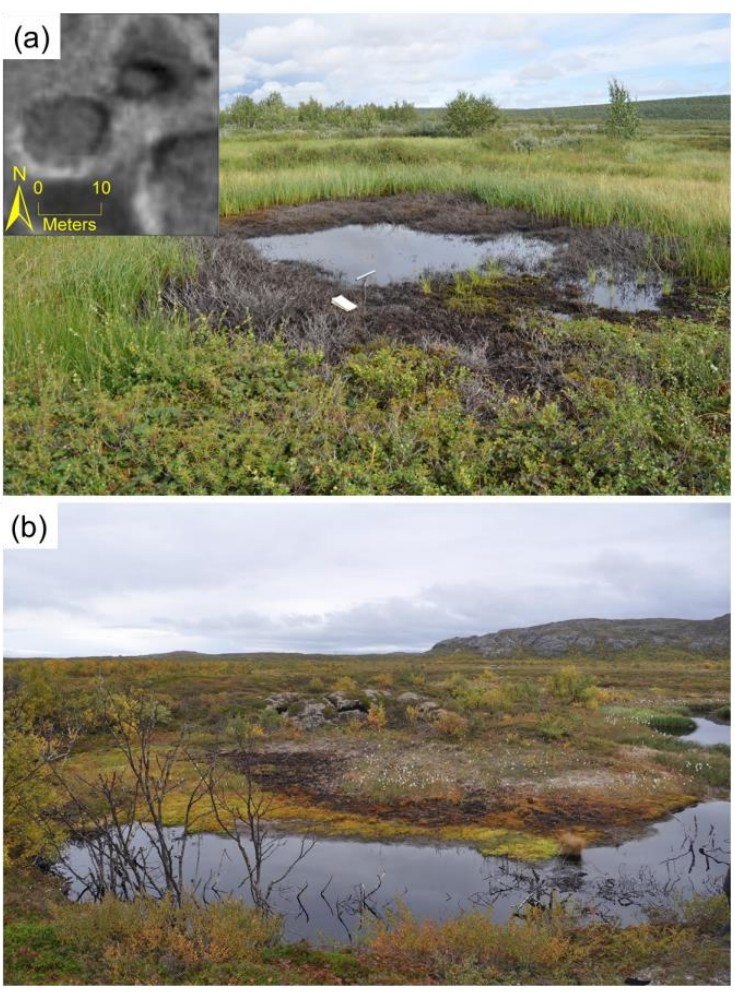

**Figure 2.** (a) Small thermokarst lake with recently dead vegetation at palsa mire 7 in *Suossjavri,* August 2014, situated at the location of a former small palsa (visible in aerial image from 2003, see inlet). (b) Degrading palsa and recently submerged birch trees at palsa mire 2 in *Karlebotn*, September 2015.

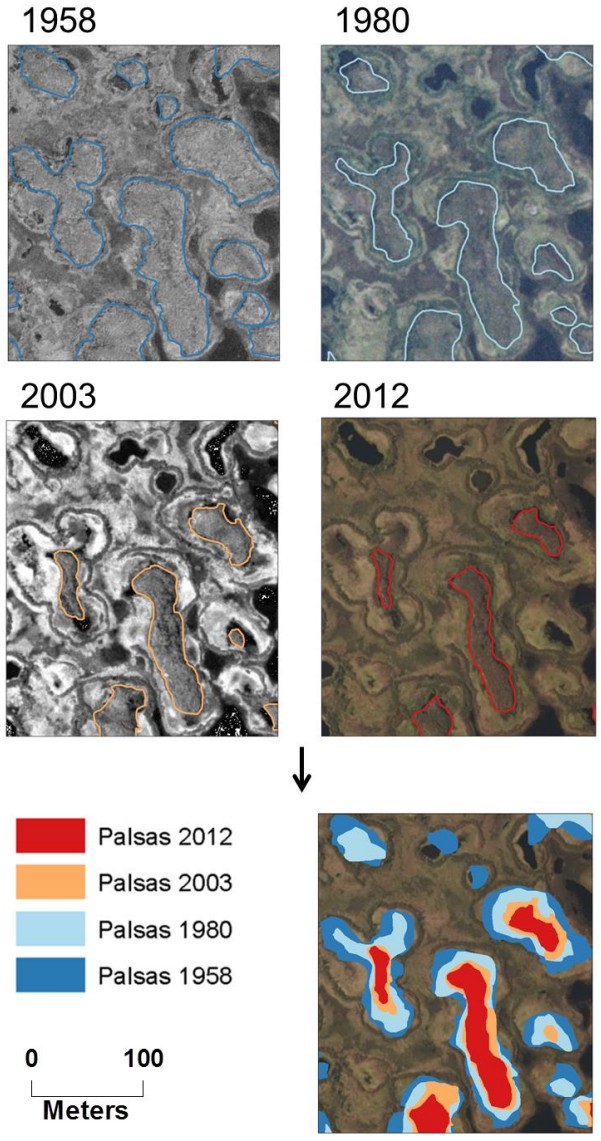

**Figure 3.** Example of polygons from the delineation of palsas at four different times in *Goatheluoppal*.

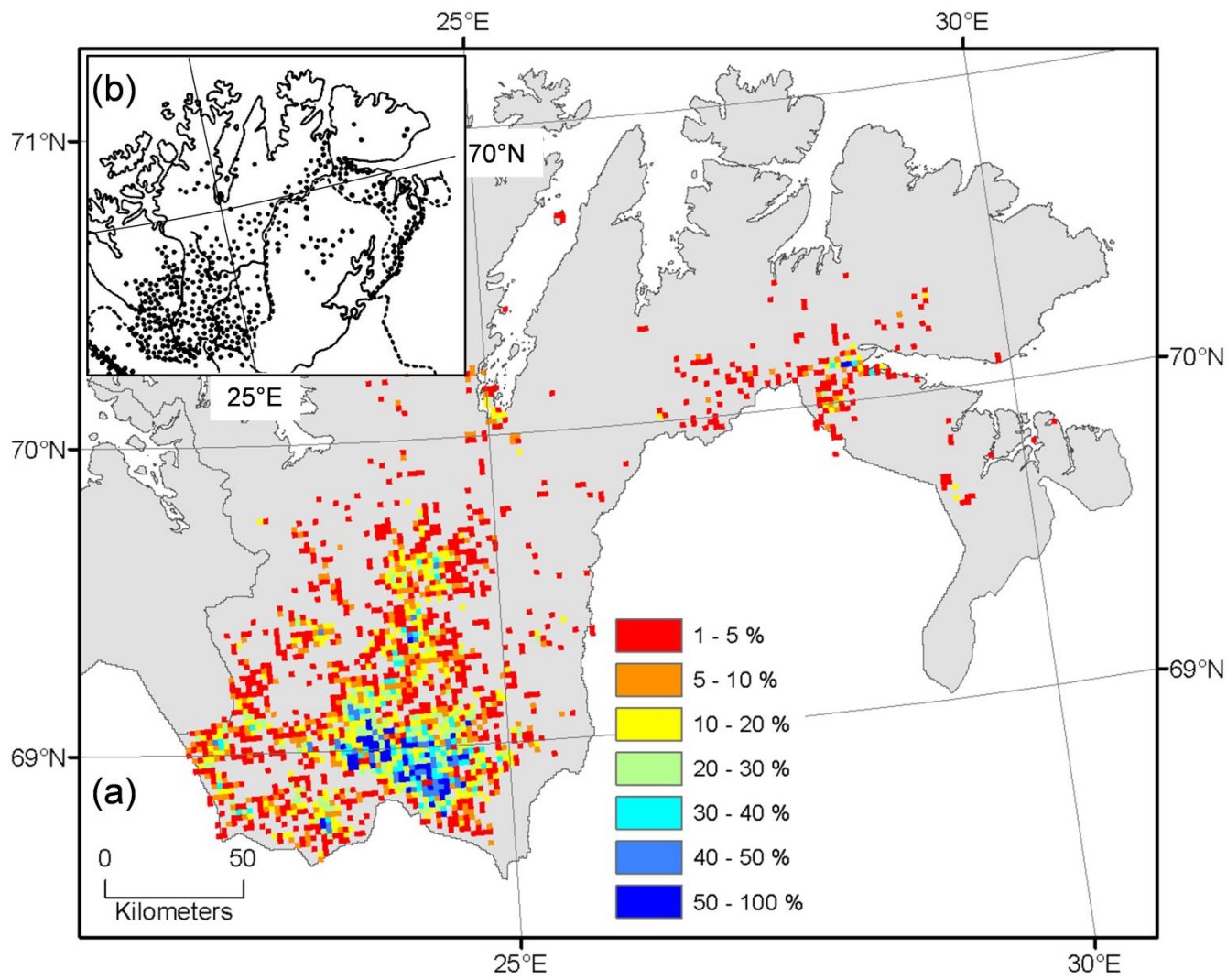

**Figure 4.** Distribution of palsas and peat plateaus in Finnmark at 2 km resolution. The concentration reflects the proportion (in percent) of 250 m$^2$ grid-cells that have presence of palsas in 2 km$^2$ grid-cells. The inlet map (b) shows the distribution of palsa mires mapped by Sollid and Sørbel (1998) based on work from the 1960s-1970s (Sollid and Sørbel, 1974).

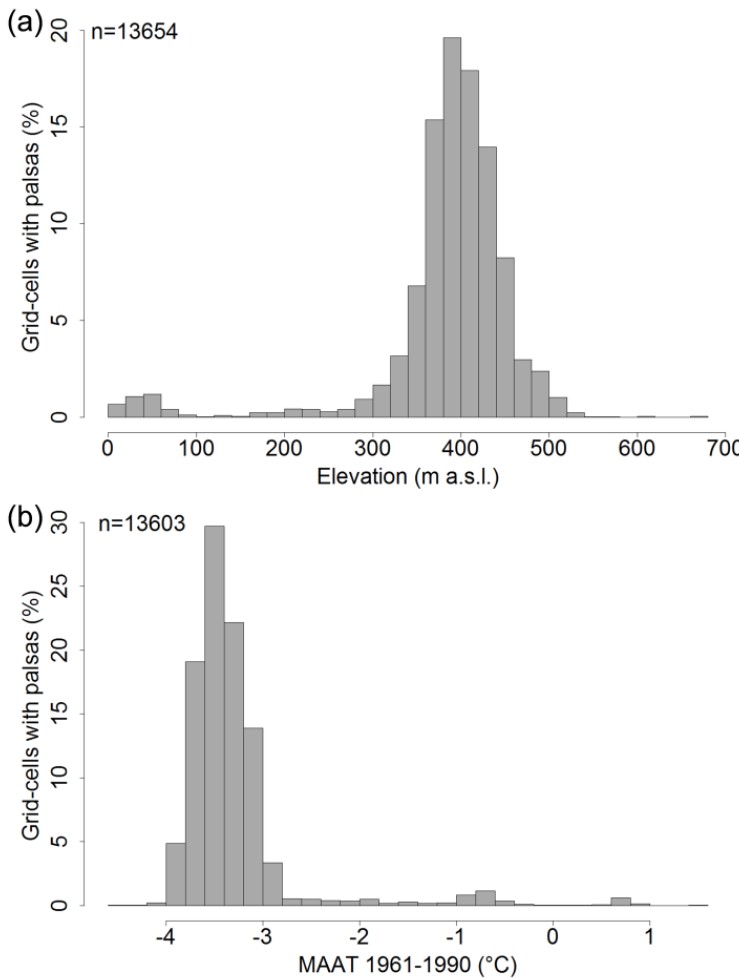

**Figure 5.** Histograms based on the location of the 250 m$^2$ grid-cells with presence of palsas and peat plateaus for (a) gridded data of the minimum elevation (Kartverket, 2015b) in each grid cell and (b) gridded data of mean annual air temperature (MAAT) for 1961-1990 (seNorge, 2016). The number of grid-cells used in the histograms deviate slightly from the mapped number of grid-cells with palsas due to insufficient covering close to the coast and to the Norwegian border of the gridded layers of MAAT and the digital elevation model.

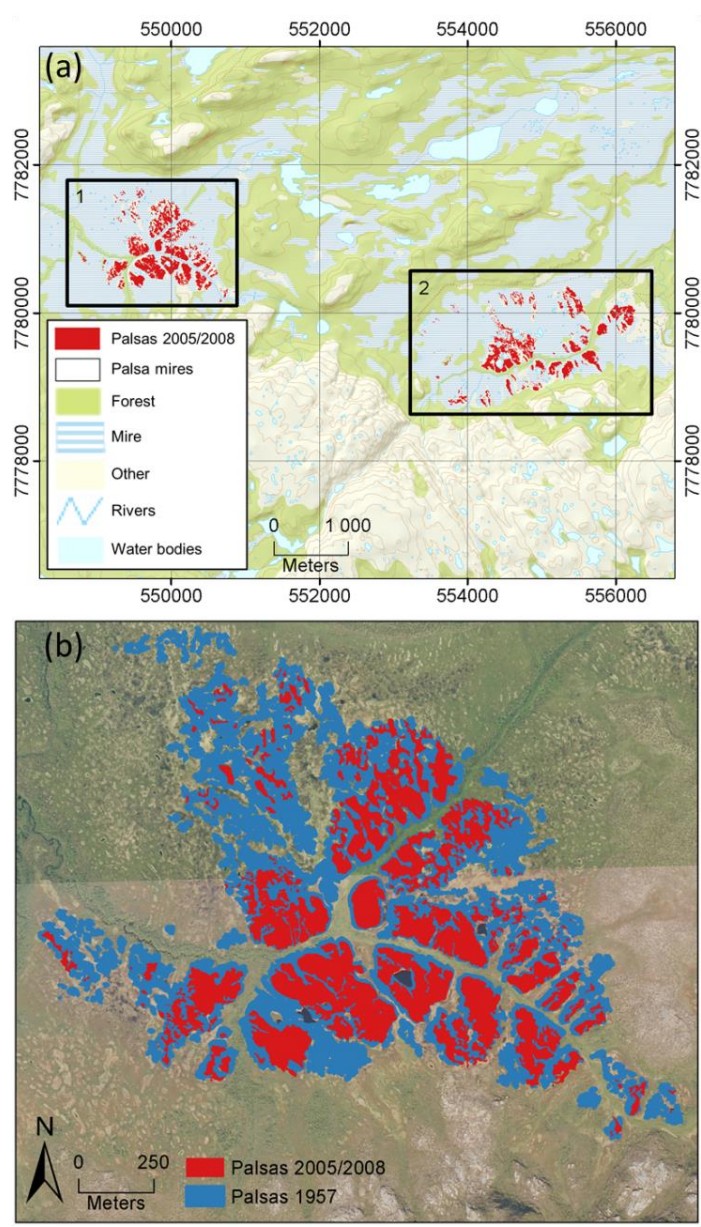

**Figure 6.** (a) Palsa mires mapped in *Karlebotn*. Background map from Kartverket (2015b) with projection in UTM 35N. (b) Palsas and peat plateaus mapped in mire 1. Background image from 2008 through Norgeibilder (2015).

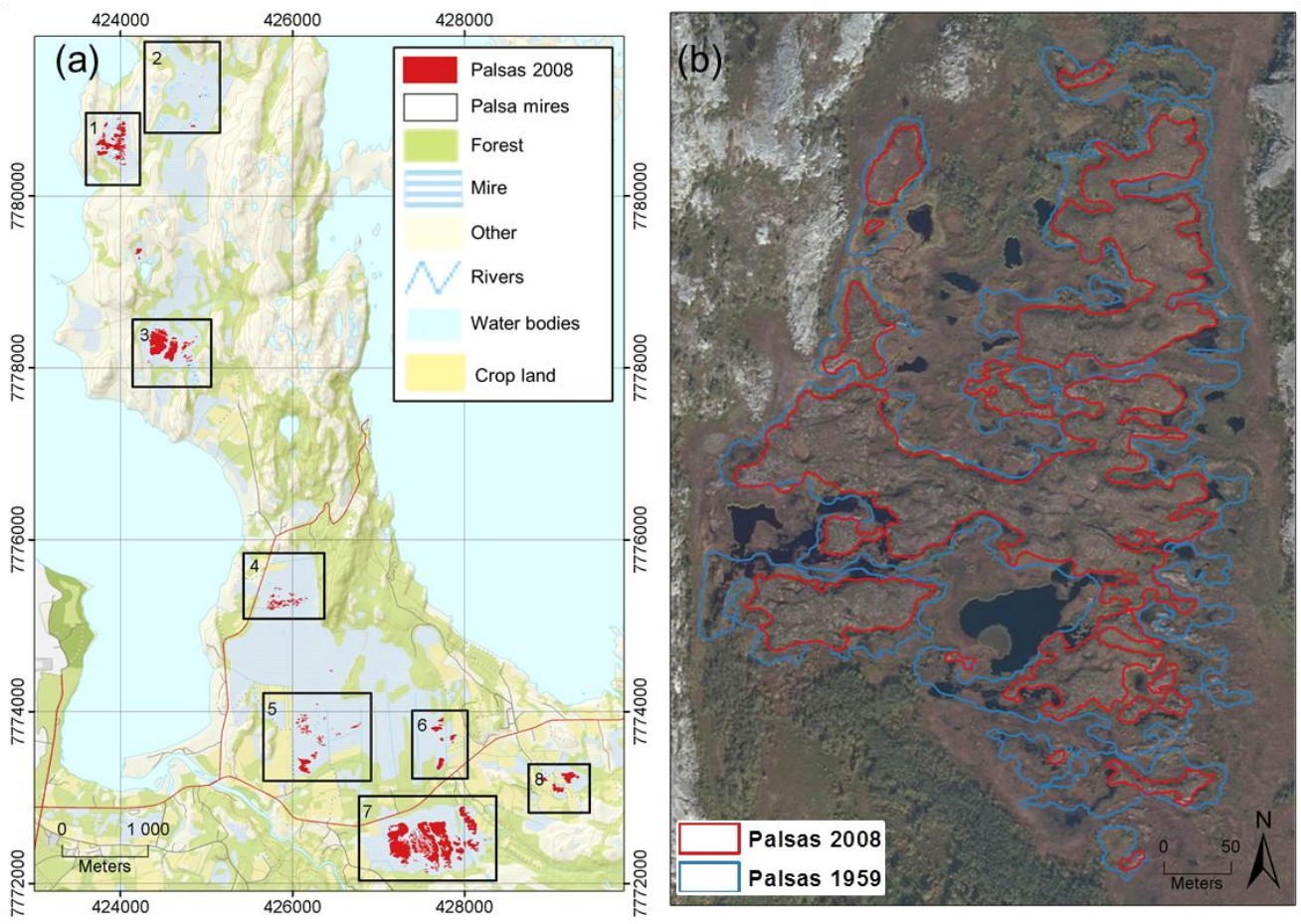

**Figure 7.** (a) Palsa mires mapped in the study area *Lakselv*. Some smaller palsas are outside these eight palsa mires. Background map from Kartverket (2015b) with projection in UTM 35N. (b) Palsas mapped in palsa mire 1 in *Lakselv*. Background image from 2008 through Norgeibilder (2015).

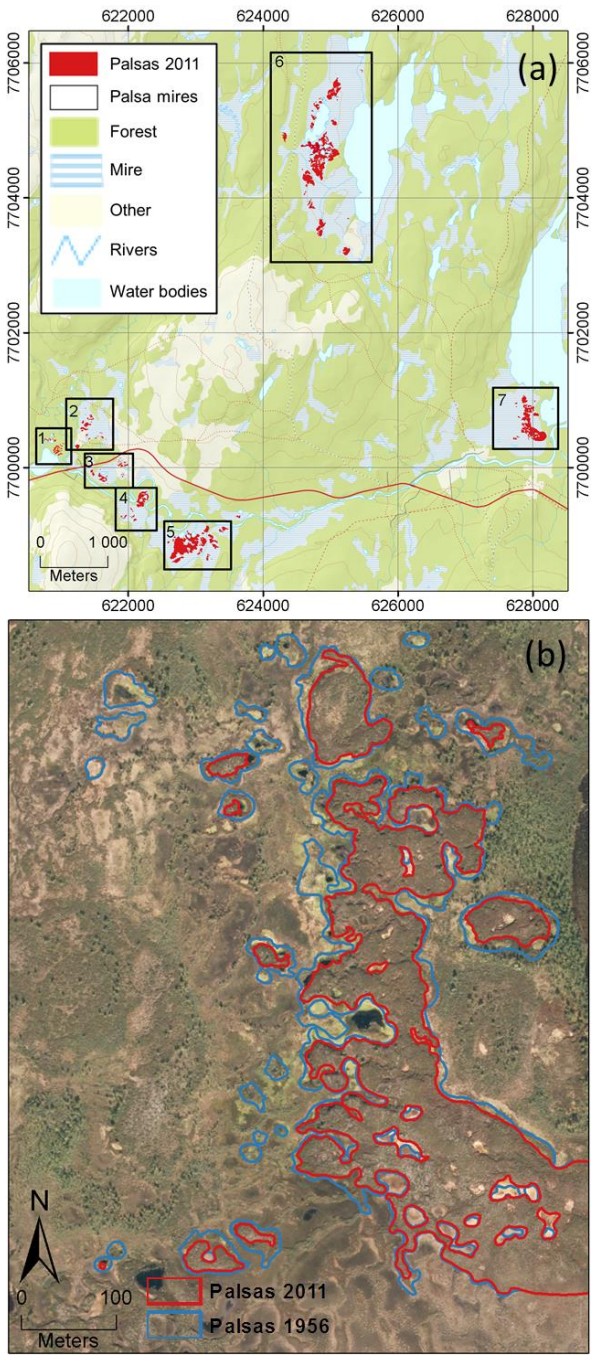

**Figure 8.** (a) Palsa mires mapped in the study area *Suossjavri*. Background map from Kartverket (2015b) with projection in UTM 34N. (b) Palsas and peat plateaus mapped in palsa mire 7 in *Suossjavri*. To increase the visibility of this figure, only the delineation of palsas from 1956 and 2011 is included. Background image from 2011 through Norgeibilder (2015).

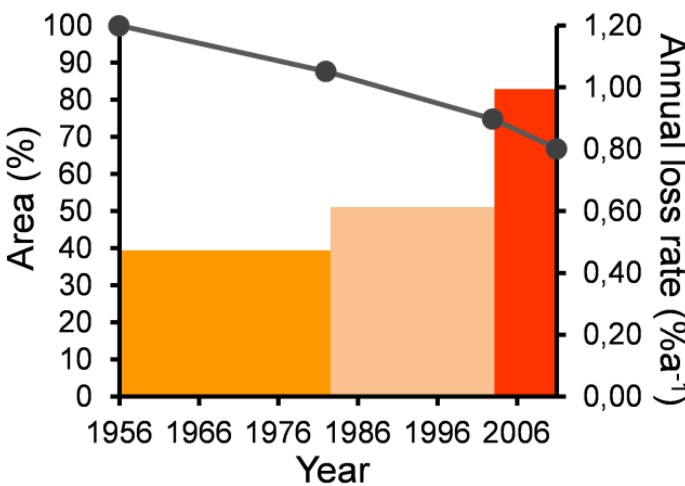

**Figure 9.** Areal extent of palsas and peat plateaus in *Suossjavri* relative to the area in 1956 (dots), and mean annual loss rates (bars) for the different time periods.

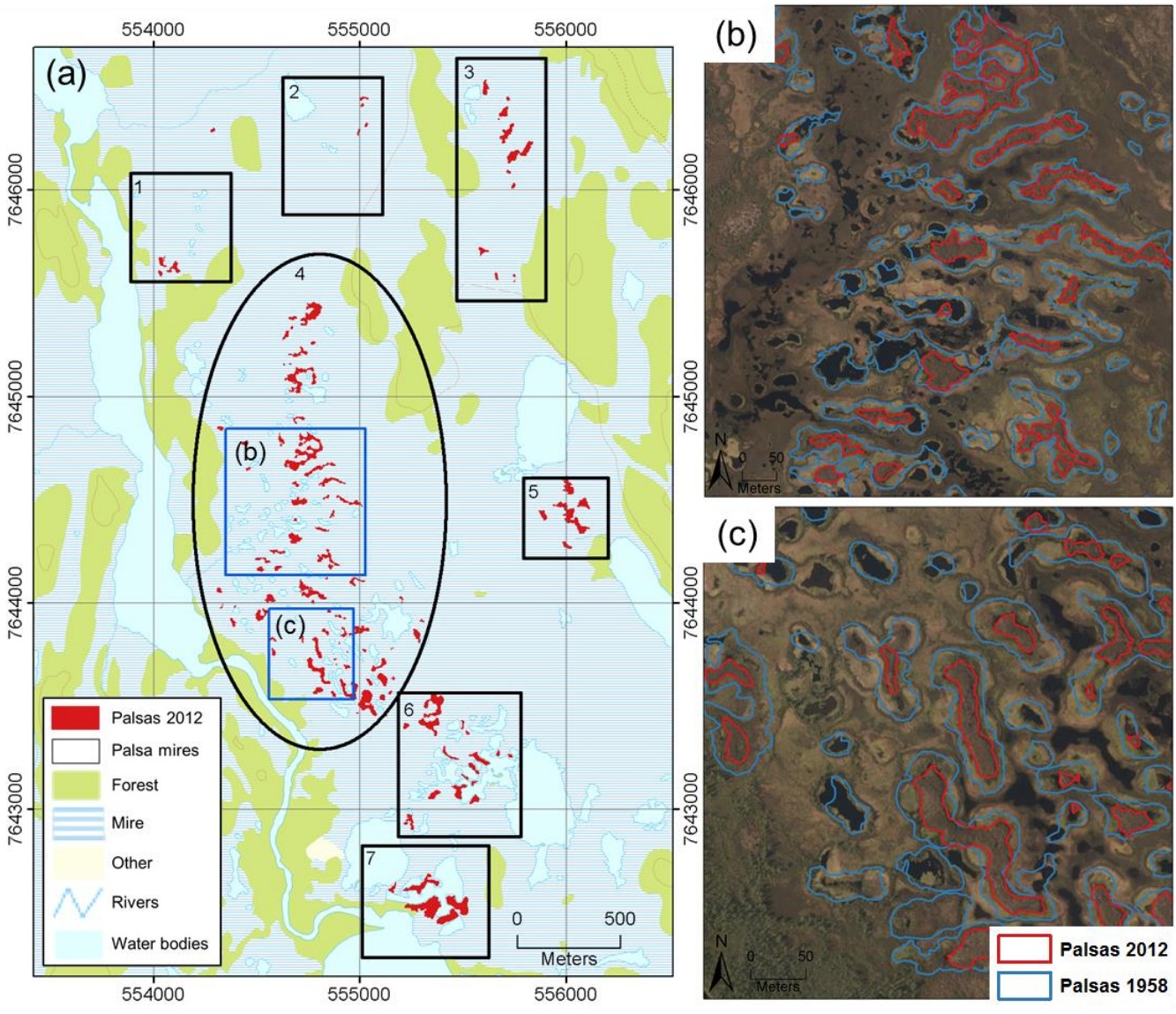

**Figure 10.** (a) Palsas mires mapped in the study area *Goatheluoppal*. Background map from Kartverket (2015b) with projection in UTM 35N. The images in (b) and (c) show examples of some palsas mapped in the palsa region 4 in *Goatheluoppal*. To increase the visibility of this figure, only the delineation of palsas from 1958 and 2012 is included. Background images from 2012 through Norgeibilder (2015).

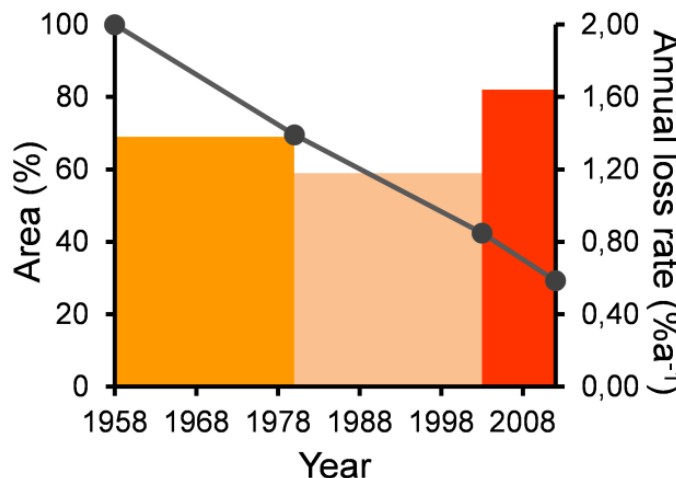

**Figure 11.** Areal extent of palsas and peat plateaus in *Goatheluoppal* relative to the area in 1958 (dots), and mean annual
15   loss rates (bars) for the different time periods.

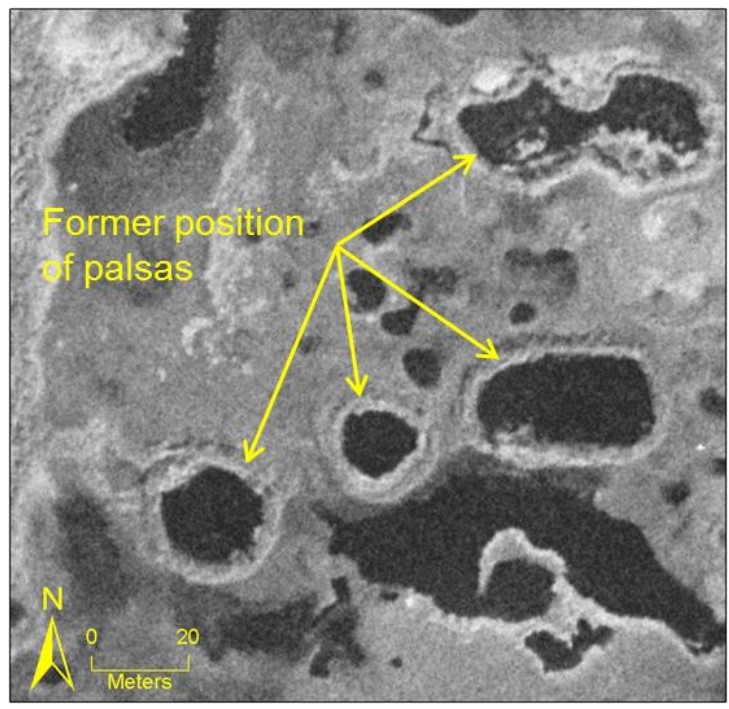

**Figure 12.** Examples of thermokarst lakes encircled by rim ridges in an image from 1958 of the palsa region 4 in *Goatheluoppal*.

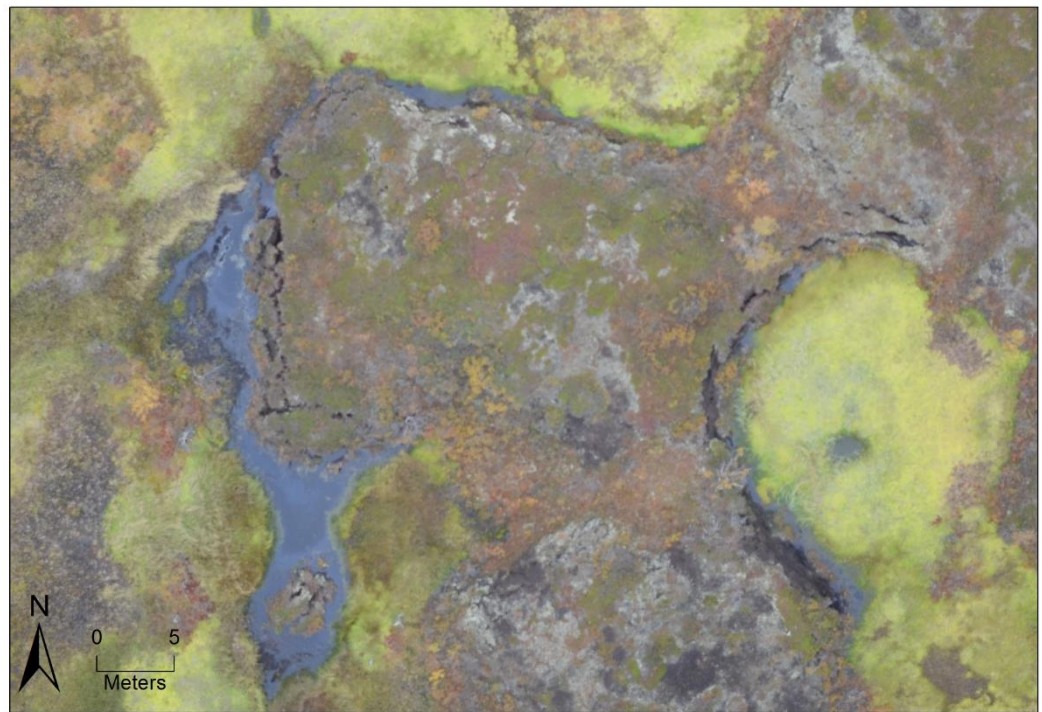

10   **Figure 13.** High-resolution aerial image of a peat plateau in mire 7 at *Suossjavri*. Note the ongoing block erosion at the margins where blocks of peat collapse in the thermokarst ponds flanking the eroding edge. The more stable margins do not feature ponds and mire vegetation grows directly at the edge of the peat plateau.