# Peer review of "Strong degradation of palsas and peat plateaus in northern Norway during the last 60 years"

_The Cryosphere, 2016_

## Short Comment (SC1) · 10 Mar 2016

There are a few more studies that I think are relevant for this paper:

About lateral erosion and numerical modeling of palsas/peat plateaus:

Kurylyk, B. L., M. Hayashi, W. L. Quinton, J. M. McKenzie, and C. I. Voss (2016), Influence of vertical and lateral heat transfer on permafrost thaw, peatland landscape transition, and groundwater flow, Water Resour. Res., 52, doi:10.1002/2015WR018057.

About palsa/peat plateau permafrost in northern Sweden:

Sjöberg, Y., P. Marklund, R. Pettersson, and S. W. Lyon (2015), Geophysical mapping of palsa peatland permafrost, The Cryosphere, 9(2), 465-478.

Åkerman, H. J., and Johansson, M.: Thawing permafrost and thicker active layers in

sub-arctic Sweden, Permafrost Periglac., 19, 279-292, 10.1002/ppp.626, 2008.

About lateral thawing of palsas/peat plateas:

Payette, S., A. Delwaide, M. Caccianiga, and M. Beauchemin (2004), Accelerated thawing of subarctic peatland permafrost over the last 50 years, Geophysical Research Letters, 31(18).
* * *

---

## Referee Comment (RC1) · Anonymous Referee #1 · 19 Aug 2016

The Cryosphere Discuss., doi:10.5194/tc-2016-12, 2016 Authors: Amund F. Borge, Sebastian Westermann, Ingvild Solheim, Bernd Etzelmüller Title: Strong degradation of palsas and peat plateaus in northern Norway during the last 60 years.

Comments The subject of this paper is appropriate to the Cryosphere Journal. The paper contains original material on importance of studies on changes (decrease in area) of palsas and peat plateaus in northern Norway since 1950s in order closer understand of possible way of the evolution of palsas and peat plateaus. Most of this material is new for the investigated area, illustrates more precisely data and analysis of this material and the results discussed in the manuscript could bring the new knowledge to the existing concept of the sequence of palsa evolution. Using high-resolution aerial imagery, authors quantified the lateral changes of the extent of palsas and peat plateaus in northern Norway. Combining the change rates with the areal mapping authors report

on widespread receding of palsas and peat plateaus area since the 1950s in northern Norway. The methodology is sound, the assumptions and objectives are clearly identified. It is a good paper and the publication of this kind of paper could be timely and beneficial for researchers working in the same field, as well as for many other researchers conducting a wide spectrum of environmental studies. From my point of view, the paper in review could be published as it is, but I have a few minor comments and questions to authors before the manuscript will be published in the Cryosphere Journal.

1. Page 7, block 30. Open the acronym LIA. 2. Conclusion. The first sentence. It is not clear to me what exact high-resolution aerial images are providing? 250 m? If so, how did you estimate that "newly formed palsas of diameter of more than 10 m were not observed"? (The last sentence at the page 10 and spatial scale at the Figure 2a).

---

## Referee Comment (RC2) · Anonymous Referee #2 · 31 Oct 2016

I read through the manuscript several times with great interests. The authors have done a thorough job by documenting changes in palsas and peat plateau in northern Norway. The work will be very valuable for palsas and permafrost studies in the Arctic. I do have some concerns and suggestions about the current version of the manuscript.

Major concerns:

1). Potential error analysis: There are several places in the work which could produce substantial errors. First, the 10 m diameter threshold. By ignoring all palsas less than 10 m in diameter could produce potentially significant errors. The authors have four in-situ sites, they should their in-situ data to evaluate how much error it may bring out. Second, the authors just use one same person to delineate the boundaries of palsas for each study site. Yes, it will be very consistent but not necessarily the lowerest in

errors. To digitize any data and information from paperwork into computer, it usually requires two persons to do the same work separately, then use a program to check each other. If both agree, pass, if not, go back and check the original paper version to reduce the human error to the minimum. If it did by the same one person as stated in this study, the potential error is unknown. The authors should seriously consider the issue.

2). The authors should provide more in-situ information, such as at a specific site or a specific palsa, what is happened and/or happening? If palsas disappeared, peat materials are still there. And also geomorphologically, what it looks like when palsa is gone. I believe that not all of them become thermokasrt ponds or lakes.

3). The authors indeed provide information about MAAT, changes in air temperature and precipitation in the study sites and the region as a whole. The authors do not provide the specific vaules for the changes in air temperature and precipitation. I hope in the revised version, this imformation will be provided. The most importanty, the authors rarely mention about snowfall and snow cover data and information. In the Arctic and Subarctic discontinuous and sporadic permafrost zones, the combination of peat layer and snow cover is often more important than air temperature in terms of permafrost presence or obsence. Changes in peat layer in a short period of time (60 years as in this study) may be very unlikely, changes in snowfall and snow cover conditions may be possible. Indeed, the authors state in the text that precipitation increased, but how much is it snowfall? What is snow cover variations? etc.

4). Some concepts are confusing: degradation of palsas, lateral erosion of palsas, and disappearance of palsa: By "degradation of palsas", we may understand it refers to the processes on the way or at the end; by "disappearance of palsas", it definitely refers to the end of palsas, and by "the lateral erosion of palsas", it is not clear it refers to lateral shrinking in size or materials are transported away. This may need to be clarified.

5). Use the results from four sites to expand to the entire Finnmark, it is kind of skeptical. What is the total area of these four sites? What is the percentage fraction of the total area of these four sites to the whole Finnmark?

Some minor comments:

1). p.1, line 29 to p.2, line 1: The authors state "The permafrost temperature in palsas is thus relatively warm", the description is not precise, temperature iself cannot be warm or cold, it can be high or low. Permafrost can be warm or cold. Just a reminder.

2). p. 2, lines 19-20: same comments above.

3). p.4., the authors mentioned about winter and summer, please be specific, which months are referring to in terms of winter and summer, this is important in the Arctic and Subarctic sine the cold season is so long. Also, when you discuss about precipitation, what is the fraction of snowfall? when you discuss about changes in precipitation, what is the fraction of changes in snowfall? This information is very important for the potential readers to understand what is going on.

---

## Author Comment (AC2) · 4 Dec 2016

Reviewer 1:

We thank the reviewer for his/her comments. We have addressed the technical corrections in a revised version of the manuscript. In the following, we give a point-by-point reply to the two points raised (in bold):

**Comments The subject of this paper is appropriate to the Cryosphere Journal. The paper contains original material on importance of studies on changes (decrease in area) of palsas and peat plateaus in northern Norway since 1950s in order closer understand of possible way of the evolution of palsas and peat plateaus. Most of this material is new for the investigated area, illustrates more precisely data and analysis of this material and the results discussed in the manuscript could bring the new knowledge to the existing concept of the sequence of palsa evolution. Using high-resolution aerial imagery, authors quantified the lateral changes of the extent of palsas and peat plateaus in northern Norway. Combining the change rates with the areal mapping authors report on widespread receding of palsas and peat plateaus area since the 1950s in northern Norway. The methodology is sound, the assumptions and objectives are clearly identified. It is a good paper and the publication of this kind of paper could be timely and beneficial for researchers working in the same field, as well as for many other researchers conducting a wide spectrum of environmental studies. From my point of view, the paper in review could be published as it is, but I have a few minor comments and questions to authors before the manuscript will be published in the Cryosphere Journal.**

**1. Page 7, block 30. Open the acronym LIA.**

Done.

**2. Conclusion. The first sentence. It is not clear to me what exact high-resolution aerial images are providing? 250 m? If so, how did you estimate that "newly formed palsas of diameter of more than 10 m were not observed"? (The last sentence at the page 10 and spatial scale at the Figure 2a).**

In the revised version, we have added the spatial resolution of the aerial images to the first sentence in Sect 6, conclusion. We write: *Using high-resolution (0.2-0.5 $m^2$) aerial imagery, we systematically map the occurrence of palsas and peat plateaus on 250 m grids in the sporadic permafrost zone in northern Norway.* Using images with a spatial resolution of 0.2-0.5 m, it is generally unproblematic to visually detect palsas with a diameter approximately larger than 10 m. The second last sentence at page 10, in Sect. 6, conclusion, is reformulated to clarify that this statement is is only valid for the four study areas. We write: *Newly formed palsas of diameter of more than 10 m were not observed in the study areas.*

On behalf of the authors,

Amund F. Borge

---

## Author Response (AR1)

**Author`s response for the manuscript "Strong degradation of palsas and peat plateaus in northern Norway during the last 60 years" by Borge et al.**

We have assembled a revised version of the manuscript in which the suggestions of two reviewers are incorporated. Most importantly, we have incorporated a better quantification of the potential uncertainties of our methodology, following reviewer 2. Furthermore, we have added a new figure with an aerial image to better illustrate some of the processes. In the following, we provide the replies to the reviewers and to Short Comment 1, followed by changes to the manuscript and the revised version of this manuscript in which changes are marked in bold.

**Reply to Reviewer 1 (his/her text in bold):**

**Comments The subject of this paper is appropriate to the Cryosphere Journal. The paper contains original material on importance of studies on changes (decrease in area) of palsas and peat plateaus in northern Norway since 1950s in order closer understand of possible way of the evolution of palsas and peat plateaus. Most of this material is new for the investigated area, illustrates more precisely data and analysis of this material and the results discussed in the manuscript could bring the new knowledge to the existing concept of the sequence of palsa evolution. Using high-resolution aerial imagery, authors quantified the lateral changes of the extent of palsas and peat plateaus in northern Norway. Combining the change rates with the areal mapping authors report on widespread receding of palsas and peat plateaus area since the 1950s in northern Norway. The methodology is sound, the assumptions and objectives are clearly identified. It is a good paper and the publication of this kind of paper could be timely and beneficial for researchers working in the same field, as well as for many other researchers conducting a wide spectrum of environmental studies. From my point of view, the paper in review could be published as it is, but I have a few minor comments and questions to authors before the manuscript will be published in the Cryosphere Journal.**

**1. Page 7, block 30. Open the acronym LIA.**

Done.

**2. Conclusion. The first sentence. It is not clear to me what exact high-resolution aerial images are providing? 250 m? If so, how did you estimate that "newly formed palsas of diameter of more than 10 m were not observed"? (The last sentence at the page 10 and spatial scale at the Figure 2a).**

In the revised version, we have added the spatial resolution of the aerial images to the first sentence in Sect 6, conclusion. We write: *Using high-resolution (0.2-0.5 m$^2$) aerial imagery, we systematically map the occurrence of palsas and peat plateaus on 250 m grids in the sporadic permafrost zone in northern Norway.* Thus, it should be clear that the high-resolution images features a spatial resolution of 0.2-0.5 m, while the mapping approach is based on 250 m grids. Using images with a spatial resolution of 0.2-0.5 m, it is relatively easy (if you know what to look for) to visually detect palsas with a diameter approximately larger than 10 m. The second last sentence at page 10, in Sect. 6, conclusion, is reformulated to clarify that this sentence only is valid for the four study areas. We write: *Newly formed palsas of diameter of more than 10 m were not observed in the study areas.*

**Reply to Reviewer 2 (his/her text in bold):**

**I read through the manuscript several times with great interests. The authors have done a thorough job by documenting changes in palsas and peat plateau in northern Norway. The work will be very valuable for palsas and permafrost studies in the Arctic. I do have some concerns and suggestions about the current version of the manuscript.**

**Major concerns:**

**1). Potential error analysis: There are several places in the work which could produce substantial errors. First, the 10 m diameter threshold. By ignoring all palsas less than 10 m in diameter could produce potentially significant errors. The authors have four in-situ sites, they should their in-situ data to evaluate how much error it may bring out. Second, the authors just use one same person to delineate the boundaries of palsas for each study site. Yes, it will be very consistent but not necessarily the lowerest in errors. To digitize any data and information from paperwork into computer, it usually requires two persons to do the same work separately, then use a program to check each other. If both agree, pass, if not, go back and check the original paper version to reduce the human error to the minimum. If it did by the same one person as stated in this study, the potential error is unknown. The authors should seriously consider the issue.**

The 10 m threshold was only used in the mapping of the palsa distribution at 250m scale, not in the delineation process for the four study areas, for which also palsas with diameter of less than 10m were mapped. We have clarified this in the revised verison of the manuscript.

In the revised version, we have also addressed the potential error of using the 10 m threshold in the distribution mapping process by using, as proposed by the referee, the data from our four study sites. This is now included in section 3. We find the difference (by comparison of using and not using a 10 m threshold) in number of grid-cells to be 8 % and the estimated difference in the total area of palsas for the four study sites to be 0.16 %, the latter of which is negligible in the context of our study. In the revised evrsion, we write:

*"To determine the uncertainty induced by the 10 m threshold in the 250 m scale mapping (see above), we once again investigate the four main study areas and their surroundings (c. 140 km2). By mapping at best possible resolution, palsas with smaller diameter can be detected which facilitates estimating the number of 250 m grid-cells excluded due to the mapping threshold. We find that the number of grid-cells with presence of palsas is 8.6 % higher when including palsas and palsa remnants with diameters less than 10 m. However, the total area covered by palsas/peat plateaus increases by only 0.16 % due to the limited area of these palsas. We therefore conclude that our mapping can provide a robust estimate for the total area covered by palsas/peat plateaus, although isolated small palsas occur regulary in grid cells flagged as free of palsas/peat plateaus."*

To address the uncertainty by only using one person to delineate the boundaries of palsas and peat plateaus, one of the authors has re-mapped about 50 % of the area of palsas/peat plateaus at mire 1 in Karlebotn for the 2008 image. Comparison of the total area gives a difference in 8 % in the mapped total area between the two persons, which is substantial, but signifantly smaller than the changes in extent over time. The process is described in Sect. 3:

*"To ensure a consistent interpretation of the extent of palsas on the aerial images, the same person delineated the palsas for each individual study area. To estimate the accuracy of the manual and thus to a certain extent subjective delineation process, parts of the Karlebotn study area (~ 0.24-0.26 km2) were independently mapped (using the images from 2005) by two persons. This comparison yielded a difference of 8 % in the total area which can be considered a rough estimate for the mapping accuracy."*

In Sect. 4.2 (Results) we write:

*"We note that the reduction in areal extent is significantly larger than the estimated accuracy of the manual delineation process (8 % of the total mapped area, Sect. 3)."*

**2). The authors should provide more in-situ information, such as at a specific site or a specific palsa, what is happened and/or happening? If palsas disappeared, peat materials are still there. And also geomorphologically, what it looks like when palsa is gone. I believe that not all of them become thermokarst ponds or lakes.**

We have added a paragraph to Section 5.3 and a new figure showing an aerial image form a peat plateau near *Suossjavri* with block erosion and pond formation clearly visible. In the revised version, we write:

*"Fig. 13 shows an aerial image of a peat plateau near Suossjavri highly affected by block erosion, as common for palsas and peat plateaus in this area. At the actively degrading margins, the mire vegetation is not yet established and a water-filled depression forms, indicating that the retreat of the margin occurs at higher velocity than the regrowing of the mire vegetation. However, the water bodies become overgrown and many of them eventually disappear which is evident from both the aerial images and field observations. The proximity between the standing water and the ice-rich core of the peat plateaus and palsas most likely contributes to thermal undercutting and eventually block erosion at the margins (Kurylyk et al., 2016), but a variety of factors, such as the height of the palsa and the ground ice content can be expected to play a role for this process.*

*On the other hand, the interior of palsas and peat plateaus can also experience thaw subsidence resulting in thermokarst depressions and suprapermafrost taliks, as seen for peat plateaus in northern Sweden (Åkerman and Johansson, 2008, Sjöberg et al., 2015). Based on calculated thaw rates and an instant increase in air temperature of 2 °C, Sjöberg et al. (2015) estimated that it will take 175-260 years for the permafrost at their investigated peat plateaus to completely thaw. However, much more rapid degradation has been observed in the same region (Zuidhoff, 2002), which could be an indication that lateral erosion considerably increases the degradation rates. A recent study in south-central Alaska found that 85 % of the degradation of forested permafrost plateaus was due to lateral degradation along the margins (Jones et al., 2016). "*

**3). The authors indeed provide information about MAAT, changes in air temperature and precipitation in the study sites and the region as a whole. The authors do not provide the specific vaules for the changes in air temperature and precipitation. I hope in the revised version, this imformation will be provided. The most importantly, the authors rarely mention about snowfall and snow cover data and information. In the Arctic and Subarctic discontinuous and sporadic permafrost zones, the combination of peat layer and snow cover is often more important than air temperature in terms of permafrost presence or obsence. Changes in peat layer in a short period of time (60 years as in this study) may be very unlikely, changes in snowfall and snow cover conditions may be possible. Indeed, the authors state in the text that precipitation increased, but how much is it snowfall? What is snow cover variations? etc.**

In the revised version, we provide additional information about snow cover and snow depth for the study areas. We write in Sect. 2: *"(...)while the mean annual (hydrological year) maximum snow depth (MASD, 1971-2000) ranges from less than 50 cm on Finnmarksvidda to more than 200 cm at the outer coast (seNorge, 2016). On Finnmarksvidda, the mean annual number of days with dry snow (MADDS, 1961-1990) is generally between 150*

*and 200 (seNorge, 2016), and the mean fraction of snow of the total precipitation (MSFr, 1961-1990) is usually less than 40 % (seNorge, 2016)."*

In Sect. 4., we write: *"MASD increased in all areas except Lakselv according to seNorge (2016) data, although it is unclear if this result is representative for palsas and peat plateaus, as snow depths on palsas/peat plateaus are generally much lower than in the surrounding wet mire due to wind redistribution, as e.g. observed in Suossjavri and Lakselv in March 2013."*

In Table 2, we have added data about maximum snow depth, days of dry snow and the mean fraction of snow for our four study areas.

In Sect. 3, we have added a short description of the snow model used by seNorge.

**4). Some concepts are confusing: degradation of palsas, lateral erosion of palsas, and disappearance of palsa: By "degradation of palsas", we may understand it refers to the processes on the way or at the end; by "disappearance of palsas", it definitely refers to the end of palsas, and by "the lateral erosion of palsas", it is not clear it refers to lateral shrinking in size or materials are transported away. This may need to be clarified.**

In the revised version, we have added a definition of the terms "degradation" (when it refers to palsas) and "lateral erosion", as it is used throughout the manuscript, in Sect. 1. We write: *"By "degradation", we refer to the processes (or the result of these processes) that decrease the volume of palsas and peat plateaus. With "lateral erosion", we mean the lateral decrease in size (as seen on 2D aerial imagery) of palsas and peat plateaus, where the margin of palsas or peat plateaus is transformed to wetland. Lateral erosion is often due to block erosion, but may also be a result of ground subsidence due to melting of excess ground ice at the edge, followed by submergence below the water table of the surrounding wet mire."*

**5). Use the results from four sites to expand to the entire Finnmark, it is kind of skeptical. What is the total area of these four sites? What is the percentage fraction of the total area of these four sites to the whole Finnmark?**

The total area of these four sites for 2010s is mentioned in Section 4.1, but we have now revised the sentence for clarity. We have also added the estimated percentage fraction of 2 % for the area of these four sites compared to the whole of Finnmark, and we also make clear that expanding the results to entuire Finnmark must be rega4rded a forst-oder approximation. In Sect. 4, we write:

*"High-resolution delineation of palsas and peat plateaus in the four study areas (for the 2010s) covered in total 260 grid cells, corresponding to about 2 % of the 250 m grid cells with palsas/peat plateaus in Finnmark. The sites cover a gradient of climatic and environmental conditions across Finnmark, so that we consider the results a plausible first-order estimate, although it is unclear if the four sites are a fully representative subsample. We find a total area of 2.13 $km^2$ within the 260 grid cells, yielding an average areal fraction of palsas/peat plateaus of about 13 % in grid cells with presence of these features. The present-day total area of palsas and peat plateaus in Finnmark can thus be estimated to about 110 $km^2$ or 0.2 % of the total land area of Finnmark, with an estimated uncertainty of 10 $km^2$ due to the manual delineation process (Sect. 3). "*

Some minor comments:

**1). p.1, line 29 to p.2, line 1: The authors state "The permafrost temperature in palsas is thus relatively warm", the description is not precise, temperature iself cannot be warm or cold, it can be high or low. Permafrost can be warm or cold. Just a reminder.**

Changed - the word temperature is removed.

**2). p. 2, lines 19-20: same comments above.**

Changed – the word warmer is replaced with higher.

**3). p.4., the authors mentioned about winter and summer, please be specific, which months are referring to in terms of winter and summer, this is important in the Arctic and Subarctic sine the cold season is so long. Also, when you discuss about precipitation, what is the fraction of snowfall? when you discuss about changes in precipitation, what is the fraction of changes in snowfall? This information is very important for the potential readers to understand what is going on.**

We have added the summer and winter months in brackets, which these data is based on. We calculated the fraction of snowfall of the total precipitation by using precipitation and snow data from seNorge. For other changes in the manuscript regarding snow depth and snow cover, see our answer above to your comment at major concern 3.

**Reply to SC1, Ylva Sjöberg (her text in bold):.**

We thank Ylva Sjöberg for her comments to our study and the additional references provided. All mentioned studies are cited in the revised version of the manuscript:

**Kurylyk, B. L., M. Hayashi, W. L. Quinton, J. M. McKenzie, and C. I. Voss (2016), Influence of vertical and lateral heat transfer on permafrost thaw, peatland landscape transition, and groundwater flow, Water Resour. Res., 52, doi:10.1002/2015WR018057.**

10 This reference is now included in the new manuscript in Sect. 5.3 and 5.4.

**Sjöberg, Y., P. Marklund, R. Pettersson, and S. W. Lyon (2015), Geophysical mapping of palsa peatland permafrost, The Cryosphere, 9(2), 465-478.**

15 This reference is now included in the new manuscript in Sect. 3 and 5.3.

**Åkerman, H. J., and Johansson, M.: Thawing permafrost and thicker active layers in sub-arctic Sweden, Permafrost Periglac., 19, 279-292, 10.1002/ppp.626, 2008.**

20 This reference is now included in the new manuscript in Sect. 5.3.

**Payette, S., A. Delwaide, M. Caccianiga, and M. Beauchemin (2004), Accelerated thawing of subarctic peatland permafrost over the last 50 years, Geophysical Research Letters, 31(18).**

25 This reference is incorporated in Sect. 1 and 5.2.

**Changes to the manuscript – an overview**

**Sect. 1, Introduction**: As suggested by reviewer 2, we have added definitions for the terms "degradation" and "lateral erosion", as they are used throughout the manuscript.

**Sect. 2, Setting**: Following reviewer 2, we have added more information on climate data concerning snow depth and coverage for the region and for the specific study sites, including the fraction of snow of total precipitation.

**Sect. 3, Methodology and data**: Following reviewer 2, we have addressed the potential uncertainties of our methodology: a) we have evaluated the uncertainty of the distribution mapping due to the 10 m threshold, and b) we have provided estimates for the potential error of manual delineation of palsas by only one person. For this reason, one of the authors has re-mapped part of the Karlebotn peat plateau complex, so that two independent evaluations are available from which the uncertainty can be estimated.

Furthermore, we have added some information about the seNorge snow-model in the methodology, following reviewer 2`s request of more information about snow cover.

**Sect. 4.1, The distribution of palsas and peat plateaus in Finnmark**: Following reviewer 2, we added the information about the percentage fraction of palsa area mapped from the four study sites compared to the estimated area for the whole of Finnmark, and we added a sentence concerning the representativeness of the study sites.

**Sect. 4.2, Areal change through lateral erosion**: In this section, we have added information about the snow depth and coverage for the study sites (reviewer 2). We also added the total area of palsas and peat plateaus mapped from the four study sites using the images from the 2010s.

**Sect. 5.3, Degradation of palsas and peat plateaus through lateral erosion**: Following reviewer 2, we have provided more information about specific processes occurring at our study sites. We have included a high-resolution aerial image illustrating the formation of water bodies following degradation of the peat plateau. We also added new relevant references in response to the Short Comment by Ylva Sjöberg (SC1, 10 mars 2016).

**Sect. 5.4, Implications for permafrost modelling and mapping**: We have added a reference in response to the Short Comment by Ylva Sjöberg (SC1, 10 mars 2016).

5    **Sect. 6, Conclusion**: Following reviewer 1, some parts of the conclusion have been reformulated.

'

[revised manuscript text omitted]

---

## Author Response (AR2)

Dear Editor,

We have revised the manuscript according to the points mentioned:

**Comments to the Author:**
**The authors basically answered all questions or suggestions concerned by the two reviewers. The manuscript will not be sent back to reviewers, but there are still some minor changes which are needed:**

**1). As reviewer 2 pointed out that errors or uncertainties are very important, the authors are asked to add one small paragraph at the end of the section 5.1 to summarize all potential errors or uncertainties from this study.**

We have added a paragraph on the uncertainties, as suggested:

*The palsa distribution map of Finnmark represents all palsas/peat plateaus that are well visible in aerial images. However, isolated small palsas (with a diameter of less than 10 m) are not well recognizable so that they are not contained in the map. A more detailed assessment in the four study areas suggests that the total number of 250 m grid cells with palsas and peat plateaus may be up to 10 % higher if also isolated small palsas are included (Sect. 3). However, as these unmapped permafrost features are very small, their contribution to the total area is negligible.*

*The total area covered by palsas/peat plateaus has been computed from the gridded 250 m palsa distribution map using an average grid cell fraction that was determined by manual delineation of the palsa/peat plateau boundaries in four study areas covering about 2 % of the total number of grid cells containing palsas/peat plateaus. The manual mapping is associated with errors, e.g. by subjectively defining the palsa margins. This "human" error is estimated to be on the order of 10 % from independent mapping of two persons (Sect. 3), which can provide a rough estimate for the grid cell fraction and the hereof computed total area covered by palsas/peat plateaus. Finally, it is unclear whether the four study areas are fully representative for the entire region, although they are situated along a transect spanning a wide range of conditions under which palsas/peat plateaus occur in Finnmark.*

**2). p15, second line from the bottom: In the reply letter to the editor, the authors said that they used images from 2008, while in the text, it says using the images from 2005. Which one is correct?**

2008 is correct, this is corrected in the manuscript.

**3). p23, add "with uncertainty of 10 km2 or about 9%" after the first sentence of the first bullet point.**

Done

**4). p23, the first sentence of the second bullet: change "at all study areas" into "at the four study areas" if this is what you are talking about. Otherwsie, it can be very easily confused or misleading.**

Done

**5). p23, line 23: Is it possible that the authors provide a rough error bar for the 100 km2 decrease in area?**

The 100km2 areal decrease is a rough estimate, but it is difficult to assign a meaningful error bar to this number. The important point here is that the outlines were always mapped by the same person so the 8 % uncertainty in areal extent cannot be directly transferred to the estimate of change (although it is very important that the changes are much larger than this uncertainty of the individual mapping). Therefore, we do not provide a specific number for the uncertainty, but have changed the corresponding passage to make clear that the 100km2 areal decrease must be considered a coarse estimate only.